



# An Unmanned Aerial System (UAS) based methodology for measuring biomass burning emission factors

5  Roland Vernooij[1], Patrik Winiger[1], Martin Wooster[2,3], Tercia Strydom[4], Laurent Poulain[5], Ulrike Dusek[6], Mark Grosvenor[2], Gareth J. Roberts[7], Nick Schutgens[1], Guido R. van der Werf[1]

[1]Department of Earth Sciences, Faculty of Science, Vrije Universiteit Amsterdam, Amsterdam, The Netherlands
[2]King's College London, Environmental Monitoring and Modelling Research Group, Department of Geography, London, UK
[3]National Centre for Earth Observation (NERC), UK
10  [4]South African National Parks (SANParks), Scientific Services, Skukuza, South Africa
[5]Atmospheric Chemistry Department (ACD), Leibniz Institute for Tropospheric Research (TROPOS), Leipzig, Germany.
[6]Centre for Isotope Research (CIO), Energy and Sustainability Research Institute Groningen (ESRIG), Groningen the Netherlands
[7]Geography and Environmental Science, University of Southampton, Southhampton, UK

15  *Correspondence to*: Roland Vernooij (r.vernooij@vu.nl)

**Abstract.** Biomass burning (BB) emits large quantities of greenhouse gases (GHG) and aerosols that impact climate and adversely affect human health. Although much research has focused on quantifying BB emissions on regional to global scales, field measurements of BB emission factors (EFs) are sparse, clustered and indicate high spatio-temporal variability. EFs are generally calculated from ground- or aeroplane measurements with respective potential biases towards smouldering or flaming 20  combustion products. Unmanned aerial systems (UAS) have the potential to measure BB EFs in fresh smoke, targeting different parts of the plume at relatively low cost. We propose a light-weight UAS-based method to measure EFs for carbon monoxide (CO), carbon dioxide ($CO_2$), methane ($CH_4$) and nitrous oxide ($N_2O$), as well as $PM_{2.5}$ (TSI Sidepak AM520) and equivalent black carbon (eBC, microAeth AE51) using a combination of a sampling system with Tedlar bags which can be analysed on the ground and airborne aerosol sensors. In this study, we address the main uncertainties associated with this 25  approach (1) the degree to which taking a limited number of samples is representative for the integral smoke plume and including (2) the reliability of the lightweight aerosol sensors. This was done for prescribed burning experiments in the Kruger national park, South Africa where we compared fire-averaged EF from UAS-sampled bags for savanna fires to integrated EFs from co-located mast measurements. Both measurements matched reasonably well with linear $R^2$ ranging from 0.81 to 0.94. Both aerosol sensors are not factory calibrated for BB particles and therefore require additional calibration. In a series of smoke 30  chamber experiments, we compared the lightweight sensors to high-fidelity equipment to empirically determine specific calibration factors (CF) for measuring BB particles. For the PM mass concentration from a TSI Sidepak AM520, we found an optimal CF of 0.27, using a scanning mobility particle sizer and gravimetric reference methods, albeit that the CF varied for different vegetation fuel types. Measurements of eBC from the Aethlabs AE51 aethalometer agreed well with the multi-



wavelength aethalometer (AE33) (linear $R^2$ of 0.95 at $\lambda = 880$ nm) and the wavelength corrected Multi-Angle Absorption Photometer (MAAP, $R^2$ 0.83 measuring at $\lambda = 637$ nm). However, the high variability in observed BB mass absorption cross-section (MAC) values (5.2 ± 5.1 $m^2$ $g^{-1}$) suggested re-calibration may be required for individual fires. Overall, our results indicate that the proposed UAS setup can obtain representative BB EFs for individual savanna fires if proper correction factors

are applied and operating limitations are well understood.

## 1. Introduction

Landscape fires, also referred to as biomass burning (BB), are one of the main sources of GHG and aerosol emissions to the atmosphere (Akagi et al., 2011; Klimont et al., 2017), the latter being responsible for large uncertainties in radiative forcing Myhre et al., 2013; Saleh et al., 2014). In stable ecosystems, biogenic carbon stocks lost in the fire are replenished through

photosynthetic carbon dioxide ($CO_2$) sequestration upon regrowth (Beringer et al., 2007; Landry and Matthews, 2016). This re-uptake is thought to neutralize the long-term climate impact of $CO_2$ emissions, whereas methane ($CH_4$),  nitrous oxide ($N_2O$) and BB aerosols are potent net climate forcers. Carbon monoxide (CO) emissions affect radiative forcing indirectly through increasing the lifetime of $CH_4$, and as a precursor for $O_3$ (Myhre et al., 2013). Largely due to the fact that landscape fires emit a mixture of net warming black carbon (BC) and net cooling aerosols (i.e. organic carbon (OC) and $SO_4$), their role in the

aerosol-induced radiative forcing is not yet fully understood. BB accounts for about half of the global BC emissions (Lu et al., 2015), and savanna fires alone are responsible for roughly 40% of this BB-emitted BC (Bond et al., 2013). Although BB emissions are becoming better constrained for some trace gases, estimating fire-related aerosol emissions proves to be more difficult due to high variability in their chemical composition (Carter et al., 2019). Combined with a limited understanding of the atmospheric oxidation and secondary aerosol formation (Vakkari et al., 2014), this results in high uncertainties in global

estimates of aerosol-induced radiative forcing from BB (Bellouin et al., 2020; Bond et al., 2013; Brown et al., 2021; Carter et al., 2020).

In situ measurements of aerosol scattering and absorption in BB smoke indicate considerable variability in single-scattering albedo (SSA) (Brown et al., 2021; Liu et al., 2014), which is a key parameter governing aerosol radiative forcing (Moosmüller

and Sorensen, 2018a; Penner et al., 1992). This variability in the SSA is in part associated with different fuel type and burning conditions, and is a major uncertainty regarding the treatment of BB aerosols in climate models (Cappa et al., 2020; Saleh et al., 2014). In BB plumes, the SSA is linearly dependent on the ratio between highly light absorbing BC and brown carbon (BrC) on the one hand, and the non-absorbing fraction of OC on the other (Brown et al., 2021; Cappa et al., 2020; Pokhrel et al., 2016). BC absorbs uniformly across the visible spectrum, with a linear dependence on the wavelength (Ran et al., 2016).

BrC, on the other hand, is light absorbing OC, which is optically distinguished from BC through an increased absorption at shorter wavelength. BrC is either directly emitted during smouldering combustion or formed as secondary organic aerosol (SOA) through oligomerization and polymerization inside the plume (Moosmüller et al., 2009).




BB emission inventories are used to study the impact of fires on regional and global biogeochemical cycles. In these inventories, emissions are generally calculated based on the modelled fuel load, satellite derived burned area and field measurements of emission factors (EF) (Seiler and Crutzen, 1980). EFs, presented in gram per kilogram of dry biomass

consumed, describe the chemical breakup of fuel into gases and aerosols during a fire. If the combustion process is incomplete, a larger portion of the biomass is emitted as $CH_4$, CO and organic particles, rather than $CO_2$ which results from complete oxidation. The modified combustion efficiency (MCE), defined as the molar $CO_2$ emissions divided by the sum of the molar $CO_2$ and CO emissions (Ward and Radke, 1993), is often used as an indication of the relative contributions of flaming (high MCE) and smouldering (low MCE) combustion (Christian et al., 2003; Yokelson et al., 2013). It is usuall calculated as from

the excess $CO_2$ and CO concentrations in the biomass burning plume compared to the ambient background (Ward and Radke, 1993). The ratio between complete and incomplete combustion, and thus the MCE, is dependent on weather conditions and fuel characteristics making it both spatially and temporally variable (Surawski et al., 2015). Although biome-specific BB EFs derived from laboratory, ground-based, and aircraft *in situ* or remotely sensed trace gas and aerosol concentration measures have been reported for a large number of chemical species (Akagi et al., 2011; Andreae, 2019; Andreae and Merlet, 2001),

field measurements are typically quite sparse, clustered and show high intra-biome variability (Andreae, 2019; van Leeuwen and van der Werf, 2011). Better understanding of this BB EF variability would improve our quantification of fire emissions, and would aid understanding of the effects of future climate- and human-induced changes in fire regimes. However gaining this understanding will typically require a large number of field-based *in situ* measurements.

Field measurements of BB EF have primarily been derived using data from *in situ* sensors carried on the ground (e.g. Zhang et al., 2015; Wooster et al., 2018), or mounted on masts (e.g. Korontzi et al., 2003; Ward and Radke, 1993) or aircraft (e.g. Liu et al., 2014, 2017; Martins et al., 1998; May et al., 2014; Yokelson et al., 2007), and have also been derived from ground-based remote sensing (e.g. Meyer et al., 2012; Wooster et al., 2011). Many laboratory studies have examined EFs during experimental burns, including those looking at the characterization of BB particulate emissions (Reid et al., 2005a, Yokelson et al., 2013).

However, the applicability of these measurements to natural fires is uncertain, considering that the evolution of aerosols is highly dependent on atmospheric chemistry and dependent upon factors such as atmospheric humidity (Akagi et al., 2011; Urbanski, 2014). Fuel consumption, smoke dispersion and atmospheric photochemistry are different in wildfires compared to laboratory experiments, leading to inconsistent EFs (Liu et al., 2014, 2017; May et al., 2014; Yokelson et al., 2013). The optical and chemical properties of BB aerosols change with the ageing of the smoke (Cappa et al., 2020; Pokhrel et al., 2016; Vakkari

et al., 2014). Differences in atmospheric lifetime, hygroscopic growth, coating of soot by OC, and susceptibility to vertical and lateral transport, all complicate EF comparisons made at different points downwind (Adachi et al., 2010).

Since fire products are not uniformly distributed over the plume, EFs should ideally represent a mixture of emissions from the smouldering and flaming phases (Akagi et al., 2013; Ward and Radke, 1993). Most atmospheric models account for



photochemical processing, but not the chemical changes associated with the initial cooling of the smoke to ambient temperature. EFs are therefore ideally measured in smoke that has already cooled to ambient temperature, but not yet undergone significant photochemical processing (Akagi et al., 2011). Aircraft measurements maybe biased towards flaming emissions, since they sample lofted emissions that typically result from higher intensity combustion, (airplane), where as

ground or tower based measuremetns maybe biased towards residual smouldering combustion (RSC) sampling since the smoke from higher intensity burns is lofted out of reach. Unmanned Aerial Systems (UAS) have the potential to offer a low-cost and versatile solution for sampling different locations within a freshly emitted, dense smoke plume (Aurell et al., 2021; Vernooij et al., 2021). UAS-compatible (i.e. lightweight and energy efficient) aerosol measurement equipment, like the MicroAeth® AE51, which measures eBC, and the TSI SidePak® AM520, which measures $PM_{2.5}$ mass concentrations, have proven useful

for measuring a large variety of atmospheric pollution sources (e.g. Alas et al., 2019; Pikridas et al., 2019; Zhao et al., 2019), including for landsape fire smoke EF derivation (e.g. Zhang et al., 2015; Wooster et al., 2018). Even though these lightweight systems have their flaws (e.g. low flowrate and small measurement cells), their gradual advancement over recent years combined with their high detection range could provide valuable insights in high-concentration smoke encountered in a dense smoke plume.

Measurements of multiple carbonaceous species are required to calculate GHG and aerosol EFs through the carbon mass balance method (Yokelson et al., 2011, 2013). In this paper we describe the development of an UAS-based system to measure *in situ* mass concentrations of BC and $PM_{2.5}$ and collect bag samples to measure mixing ratios of $CO_2$, CO, $CH_4$ and $N_2O$. We focused on gases in a series of measurements of prescribed fires in the Kruger National Park (KNP), South Africa, in which

we compare UAS measurements to continuous stationary measurements on top of a mast. We then compare aerosol analysers against high-fidelity laboratory equipment to understand their limitations and calculate the specific calibration factors (CFs) for the measurement of fresh BB particles. Using the CFs from the laboratory experiments, we finally calculate the EFs for $CO_2$, CO, $CH_4$, $N_2O$, BC and $PM_{2.5}$ for the fires sampled in KNP.

## 2. Methods

We assessed the performance of the measurement system in two phases. In the first phase, we compared the UAS gas samples to mast measurements (an established methodology), during a series of prescribed burning experiments. In the second phase, we tested the performance of the individual measurements in a series of laboratory fire experiments, using the reference methods listed in Table 2. The GHG were measured using cavity ring-down spectroscopy (CRDS), with the analysers calibrated using certified standard gases at dense plume concentrations (listed in Table 2). Since the stability of GHG sampling

in Tedlar® bags has been demonstrated (Alves et al., 2010; Meyer et al., 2012; Silva et al., 2016) and CRDS is well-established, our focus in the second phase was on the measurement of $PM_{2.5}$ and eBC using UAS-compatible equipment.





## 2.1 Field measurements of GHG emission factors

### 2.1.1. Fire experiments in the Kruger National Park

We performed prescribed burning experiments throughout four fieldwork campaigns in the KNP in South Africa. The mean annual rainfall in the KNP varies from around 350 mm year[-1] in the north to approximately 750 mm year[-1] in the south and is mostly concentrated in the months November−April (Venter and Gertenbach, 1986). Although some fires start as early as March, the peak of the fire season occurs between August and October. Prescribed experimental fires on the roughly 100 by 200-meter experimental burn plots (EBPs) spread over four major vegetation types of the KNP, started over 65 years ago (Van der Schijff, 1954), and the experiments spanning this period are elaborately described by Biggs et al. (2003). KNP has an average fire interval ranging from 2−5 years, with more frequent burning towards the high-rainfall areas. The EBPs themselves are protected by fire breaks and managed at fire-frequencies ranging from 1-6 years. Besides the burning frequency, the time of treatment is varied; i.e. different plots are burned in February, April, August, October or December. Table 1 lists the total number of experiments and the spread over the different vegetation types and burn months.

### 2.1.2 Mast and UAS measurement setup in the KNP experiments

We measured BB smoke mixing ratios of different compounds using a 15-meter telescopic mast – which was located *in situ* during passage of a fire front. A box containing the CRDS gas analysers (equipment listed in Table 2) was dug in on the leeward side of the mast. The gas inlet at the top of the mast was fitted with a sintered 60 μm filter and a continuous flow (4 L min[-1]) of smoke was transported through a polyurethane tube to the gas analysers. Using a nafion gas-dryer (MD Series™, PermaPure), $H_2O_{(g)}$ was stripped from the sample stream to prevent condensation. We used a TSI SidePak™ AM520 Optical Particle Counter (OPC) (hereafter called "AM520") to measure aerosol mass concentrations (<2.5μm) and a lightweight aethalometer (MicroAeth® AE51, Magee Scientific hereafter called "AE51") to measure the eBC fraction of the PM mass concentration. A platform mounted at the top of the mast held the AM520, AE51 and inlet of the gas sampling tube. The flowrate of the AE51 was set to 50 mL min[-1], and new filter strips were installed before every fire to minimize the effect of nonlinearity in the filter-loading. Mast measurements were started about 15 minutes before fire ignition on the upwind side. The average atmospheric mixing ratio of those 15 minutes was used as the background.

For UAS sampling, we used a Matrice 100 (DJI) that filled single-polypropylene fitted bags (Tedlar® type 232-01, SKC) with smoke. Each flight, four bags are filled for 35 seconds each, at an altitude of roughly 15 meters. The methodology for the bag sampling and subsequent measurement is described in detail in Vernooij et al. (2021). Samples were taken within 3m of the mast, while preventing the disturbance of the airflow around the mast by the propellers. Bag samples were kept away from UV-radiation and analysed within 12 hours of sampling. The gas analysers were calibrated between the campaigns and we included sample bags of standard gas in each analysis session to determine analyser drift.




## 2.2 EF calculations

The excess mixing ratios (EMR, sample minus background concentrations) of the GHG and aerosols were converted to EFs using the carbon mass balance method (Yokelson et al., 1999):

$$EF_i = F_c \times \frac{MW_i}{AM_c} \times \frac{C_i}{C_{total}} \qquad (1)$$

where $EF_i$ is the emission factor of species $i$ (usually reported in g kg$^{-1}$) and $Fc$ is the fractional carbon content of the fuel by weight (estimated at 50% following Akagi et al., 2011). $MW_i$ is the molecular weight of species $i$ which is divided by the atomic mass of carbon, $AM_c$. $C_i$ is the moles of carbon per mole of species $i$ multiplied by the EMR of species $i$. $C_{total}$ is the total number of moles of emitted carbon in all carbonaceous species. Because we did not measure the non-methane hydrocarbons and the chemical composition of carbonaceous particulates, these fractions were estimated based on literature values in order to estimate $C_{total}$. For the bag and mast measurements, we used the PM to CO ratio based on AM520 and CRDS measurements, with carbon accounting for 68% of the PM-mass (Reid et al., 2005a). On average, the PM to CO ratio in our measurements was $0.0946 \pm 0.0218$ which corresponds well with the $0.0969 \pm 0.0403$ average for savanna fires (Andreae, 2019). We calculated the EFs for eBC and PM$_{2.5}$ based on the emission ratio ($ER_{(\frac{i}{y})}$) of the species $i$ to a relatively inert, co-emitted carbon-containing species $y$ following Eq. (2).

$$EF_i = ER_{(\frac{i}{y})} \times \frac{MV_y}{MW_y} \times EF_y \qquad (2)$$

The total amount of carbon in non-methane hydrocarbons was estimated to be 3.5 times the ER(CH$_4$/CO$_2$) based on common ratios for savanna fires (Andreae, 2019; Yokelson et al., 2011, 2013). To convert parts per million to mg m$^{-3}$, the reference gas emissions were multiplied by the molar volume ($MV_y$) divided by the molecular weight of the reference species ($MW_y$). For both eBC and PM$_{2.5}$, we used CO as a reference gas. We calculated fire-averaged EFs based on the cumulative EMR of the respective trace gas species in all the fire samples, causing samples with low overall trace gas concentrations to have lower impact on the fire-averaged EF.

## 2.3 Aerosol calibration experiments

For the second phase, we performed BB experiments in the Leipzig aerosol chamber at the Leibniz Institute for tropospheric research (TROPOS), the Kings Wildfire Testing Chamber (KWTC) in London, and the Fire Laboratory of Amsterdam for Research in Ecology (FLARE) where we calibrated the mobile aerosol analysers against different types of high-fidelity laboratory equipment. At TROPOS and FLARE, wooden logs, wood chips, and hay were burned in an actively vented combustion chamber, connected to the measurement equipment. At the KWTC, smoke from peat fires and straw was allowed


to stabilize around predetermined levels in a smoke chamber which was connected to a series of analysers. Table 2 lists the analysis equipment used for the different measurements.

Although many studies have measured BC EFs (Andreae, 2019), there is still much inconsistency in the precise terminology
and symbology used concerning BC measurements. In this study we follow terminology recommendations by Petzold et al. (2013), i.e. equivalent black carbon (eBC) refers to BC measured by optical absorption methods (e.g. aethalometers and MAAP), whereas elemental carbon (EC) refers to the thermally derived BC fraction, with optical correction for OC pyrolysis. Symbology may therefore be different from other papers (e.g. in this paper, $\sigma$ refers to the absorption coefficient rather than mass absorption cross-section (MAC)).

**2.3.1 eBC measurements**

The AE51 measures the rate of change in the absorption of transmitted light ($\lambda$ = 880nm) due to the continuous collection of aerosol deposits on a Teflon-coated T60 borosilicate glass-fiber filter. The measured spot is compared with a reference spot on the filter, and the eBC mass concentration is calculated through the attenuation (ATN) of the laser transmittance. The main uncertainties regarding filter-absorption eBC measurements like this are related to the calibration factor (C), which is applied
because, compared to the airborne state, the path of light is increased in the filter material by multiple scattering, resulting in enhanced optical absorption of the deposited particles and the assumed MAC (Drinovec et al., 2015; Kumar et al., 2018). C can be determined through cross-calibration with other absorptivity measurements, whereas the MAC value can be derived using co-located thermo-optical EC analysis (Gundel et al., 1984; Kumar et al., 2018). Absorptivity measurements were compared to the Multi-Angle Absorption Photometer (MAAP; type 5012, Thermo-Fisher Scientific) and a multi-wavelength
aethalometer (AE33, Magee Scientific). The latter uses the same operating principle as the AE51, but has a much higher flow rate of 6 L min$^{-1}$ and uses a real-time loading-effect (i.e. the non-linear increase in light absorption with increased loading) compensation algorithm (Drinovec et al., 2015). Its quartz-fibre filter tape automatically advances when ATN at 370 nm exceeds a threshold, further minimizing the abovementioned loading effect. The MAAP measures the transmittance and reflectance at multiple angles using the two-stream approximation in the radiative transfer scheme (Petzold et al., 2013). During
the high-concentration measurements in BB smoke, we set the flow rate to 50 mL min$^{-1}$. Although the aethalometers and the MAAP respectively measure attenuation coefficient and absorption coefficient at different wavelengths, both instruments report a mass concentration of eBC ($\mu$g m$^{-3}$). Understanding the nuances in this conversion is crucial in interpretation of the data.

**2.3.2 Absorption comparison at multiple wavelengths**

To determine the significance of the non-linearity effect of the loading, the C-value and the light absorption due to BrC, we compared the AE51 to the AE33 and the MAAP. For absorption intercomparison at different measurement wavelengths ($\lambda$),





the mass concentration of BC, presented by the aethalometers was converted to the absorption coefficient ($\sigma_{Ap}$) through the attenuation coefficient ($\sigma_{ATN}$) using Eq. (3) and (4).

$$\sigma_{ATN} = BC \times MAC_{filter} \tag{3}$$

$$\sigma_{AP} = \frac{\sigma_{ATN}}{C \times R_{ATN}} \tag{4}$$

where $MAC_{filter}$ is the spectral mass-specific attenuation cross-section for particles loaded on a filter (Müller et al., 2011) and $R_{ATN}$ stands for the Weingartner correction factor (Weingartner et al. (2003)) and is explained in more detail in Eq. (5) - (7).

10 The $MAC_{filter}$ values of 12.2 m$^2$ g$^{-1}$ for the AE51 and 14675 / $\lambda$ m$^2$ g$^{-1}$ for the AE33 (Drinovec et al., 2015; Gundel et al., 1984) were used to convert absorption into BC mass concentrations and are fixed in the analyser firmware. In Eq. (4) the optical enhancement factor C has pre-determined values of 1.56 (for AE51 Teflon-coated glass-fiber filters) and 2.14 (for AE33 quartz-fiber filter tape). Aside from the filter material used, the C-value also depends on the particle type. Through comparison with different absorptivity analysers, we determined whether the assumed C-value of the AE51 should be adjusted for BB

15 measurements

The Weingartner correction factor is calculated as follows:

$$R_{ATN} = \frac{1}{f-1} \frac{lnln\,(ATN) - lnln\,(10)}{-lnln\,(10)} + 1 \tag{5}$$

In which $R_{ATN}$ is a factor to compensate for increasing filter load, also known as the loading or shadowing effect, and $f$ is a fit parameter proportional to the absorptivity of the particulate matter (Petzold et al., 1997):

$$f = a(1 - \omega_0) + 1 \tag{6}$$

In Eq. (6), $a$ is an empirically determined parameter. Based on extrapolation of measurements by Weingartner et al. (2003) we assumed $a$ to be equal to 0.83 at 880nm. $\omega_0$ is the single scattering albedo. As we did not measure the scattering coefficient, we estimated $\omega_0$ using the empirically determined relation described by Pokhrel et al. (2016), see Eq. (7).

30 $$\omega_0 = 0.99 - 1.07 \times \left(\frac{BC}{TC}\right) \tag{7}$$





BC is the experiment-averaged eBC mass-concentration measured by the AE51 and TC is the experiment-averaged PM from the AM520 multiplied by the average carbon mass percentage of BB particles (68%, Reid et al., 2005a). Using this method, we found a mean $\omega_0$ of $0.84 \pm 0.08$, closely matching $\omega_0$ values measured for biomass burning in field experiments (Eck et al., 2013; Reid et al., 2005b). We used Eq. (8) to calculate $\sigma_{AP\ MAAB}^{637nm}$ from the reported eBC mass concentration measured by the MAAP .

$$\sigma_{AP\ MAAB}^{637nm} = eBC \times MAC_{air} \times 1.05 \tag{8}$$

$MAC_{air}$ is the mass absorption coefficient of $6.6$ m$^2$ g$^{-1}$ used in the MAAP firmware and $1.05$ is a factor to correct for the actual wavelength of the MAAP light source, which is 637 nm instead of the 670 nm assumed by the MAAP firmware (Müller et al., 2011). The spectral dependence of aerosol absorption is usually described by a power-law relationship and parameterized as $(\lambda)^{-Å_{ap}}$, where $\lambda$ is the wavelength and $Å_{ap}$ is the absorption Ångström exponent. The AE51 only measures ATN at a single wavelength in the near-infrared at $\lambda = 880$ nm, which is often used to detect absorption by BC (Drinovec et al., 2015). At this wavelength, BrC has a MAC value in the range of 0.4-0.6 m$^2$ g$^{-1}$ against 7.8 m$^2$ g$^{-1}$ for BC (Sandradewi et al., 2008; Yang et al., 2009). The AE51 therefore assumes absorption by non-BC aerosols to be negligible at this wavelength (Ran et al., 2016), meaning the signal ATN is attributed to BC absorption alone. For comparison between the MAAP and the AE51, we used Eq. (9) to calculate the AE51 absorption coefficient at 637 nm.

$$\sigma_{AP\ AE51}^{637nm} = \sigma_{AP}^{880nm} \left(\frac{637}{880}\right)^{-Å_{ap}} \tag{9}$$

In Eq. (9), the $Å_{ap}$ was determined following Eq. (10) using the 660 and 880 nm channels of the AE33 aethalometer.

$$Å_{ap}(660,880) = \frac{ln\left(\frac{\sigma_{ap}(660)}{\sigma_{ap}(880)}\right)}{ln\left(\frac{660}{880}\right)} \tag{10}$$

For BC, $Å_{ap}$ is at unity, meaning absorption dependence on wavelength is linear. Any relationship with $Å_{ap} > 1.0$ is interpreted as BrC. $Å_{ap}(660,880)$ was used to correct for the absorption contribution of BrC to the MAAP absorption coefficient (Kumar et al., 2018). Using the different wavelengths of the AE33 aethalometer, we calculated the contribution of BrC light-absorbing species. $Å_{ap}$ for the total PM was determined for each fire experiment by fitting an exponential curve through the fire-integrated, wavelength-specific absorption coefficients, derived from the multi-wavelength AE33 aethalometer. The separate Ångström exponent for BrC absorption ($Å_{ap,Brc}$) was calculated using Eq. (11) (Ran et al., 2016):


$$\sigma_{AP}(\lambda) = \sigma_{AP,BC}(\lambda_0) \times \left(\frac{\lambda}{\lambda_0}\right)^{-1} + \sigma_{AP,BrC}(\lambda_0) \times \left(\frac{\lambda}{\lambda_0}\right)^{-\text{Å}_{ap,Brc}} \tag{11}$$

Where $\sigma_{AP}(\lambda)$ is the measured absorption coefficient at wavelength $(\lambda)$, $\lambda_0$ is the reference wavelength (880 nm) and $\sigma_{AP,BC}(\lambda_0)$ and $\sigma_{AP,BrC}(\lambda_0)$ are the black- and brown carbon absorption coefficient respectively, at the reference wavelength $(\lambda_0)$.

### 2.3.3 EC/OC analysis using Sunset analyser

Unlike BC measurements thermal-optical measurements of EC are not susceptible to unertainties related to a fixed MAC value and are therefore used for aethalometer calibration (Gundel et al., 1984; Kumar et al., 2018; Salako et al., 2012). By equating EC filter measurements to eBC absorption coefficients from the AE51 we determined the "actual" MAC values in the measured BB smoke, and compared this to the MAC value of 7.8 $m^2$ $g^{-1}$ assumed by the firmware. The MAC can be calculated based on the EC value of the filter and integration of the collocated absorption coefficient measured by the AE51 following Eq. (12) (Kumar et al., 2018; McClure et al., 2020) over the time it took to load the filter.

$$MAC = \frac{\int(\sigma_{AP})_{AE51}}{EC_{filter}} \tag{12}$$

During the experiments made at KNP, TROPOS, and experiments performed at the FLARE lab, we loaded pre-fired (800ºC, 48h) 37mm quartz-fiber filters with smoke at a flowrate of 3 L $min^{-1}$ for the duration of the fire. The filters were analysed at the Centre for Isotope Research, Groningen University using an OC-EC Aerosol Analyzer (Sunset Laboratory Inc.) with a using non-dispersive infrared spectroscopy detector. The distinction between OC and EC was based on the EUSAAR_2 protocol (Cavalli and Putaud, 2008) using the transmittance of a laser beam ($\lambda = 630$ nm) through the filter to determine the relative contribution of OC and EC of the measured filter (Bauer et al., 2009). The measurement setup and measurement protocol are described in detail by Zenker et al. (2020).

### 2.3.4 PM mass concentration and size distribution

The AM520 is an optical particle counter (OPC) that uses 90° light scattering of a laser diode at a wavelength of 650 nm and has a size measurement range of 100 nm to 10 µm. It is factory-calibrated against the respirable fraction (<4.0µm) of standard ISO12103-1 (Arizona Road Dust) aerosols with a density of 2.65 mg $m^{-3}$ and a volumetric mean diameter (VMD) of 2.12 µm (Jiang et al., 2011). Additional (re)calibration is therefore needed to account for the different characteristics (e.g particle density and size distribution) of BB aerosols. The AM520 uses a linear calibration factor (CF) to convert the Arizona Road Dust (CF = 1.0) mass concentration to the desired aerosol type, which is empirically determined using Eq. (13):


$$CF_{new} = \frac{PM\,ref\,(mg\,m^{-3})}{PM\,AM520\,(mg\,m^{-3})} \times CF_{old} \qquad (13)$$

where $PM\,AM520$ is the concentration measured by the AM520, and $PM\,ref$ is the reference concentration. At TROPOS, we simultaneously measured diluted smoke with the AM520 using a inertial impactor with a cut-off of 1.0 μm, and a Mobility Particle Size Spectrometer (TROPOS-Type MPSS, (Wiedensohler et al., 2012)) with an electrical mobility size range of 0.03-0.80 μm. To match the range of the AM520, we assumed a log-normal particle size distribution and extrapolated the particle number concentration from the SMPS, as described by Heintzenberg (1994). We used Eq. (14) to calculate the particle volume of each electrical mobility diameter bin from the size distribution. Assuming spherical particles with a dynamic shape factor of unity, the electrical mobility diameter equals the geometrical diameter.

$$PM_{1.0} = \sum_{i=0.1}^{n} \frac{4}{3}\pi\left(\frac{D_{bin}}{2000}\right)^3 \times \frac{Counts_{bin}}{Air\,Volume_{bin}} \times \rho_{eff} \qquad (14)$$

Where $D_{bin}$ is the mean mobility diameter of the bin and $\rho_{eff}$ is the effective density of the particles. The sum of all masses for classes in range 0.1-1 μm was then compared to the measured < 1μm fraction from the AM520. We converted the total volume of particles for each size class to mg m$^{-3}$ assuming an effective density of $\rho_{eff}$ = 1.50 g cm$^{-3}$ which is typical for wood burning (Kumar et al., 2018; Moosmüller et al., 2009).

**2.3.5 Gravimetric analysis**

At the Kings Wildfire Testing Chamber (KWTC), London we performed calibrations of a set of co-located measurement equipment using tropical peat (from Kalimantan) and straw fuels. Smoke from the fires was collected in an approximately 3 × 3 × 3 m size sampling chamber into which the co-located measurement equipment was placed. To generate a gravimetric calibration curve, the smoke concentration was kept stable for roughly 1 hour at 100 μg m$^{-3}$ intervals ranging from 200-600 μg m$^{-3}$. The co-located equipment included six AM520s and two EA51s. Reference equipment which had their inlets sampling from the same smoke chamber were a Tapered Element Oscillating Microbalance (TEOM1400, ThermoFisher scientific), a particulate sampler (Partisol 2000i, ThermoFisher scientific) and a 37mm filter (Tissuquartz 2500QAT-UP, Merck) sampler (Personal Modular Impactor, SKC) for EC and OC analysis.

**3. Results**

We first discuss how lightweight UAS-based measurements and those from a mast-setup compared, and how the UAS measurements can be used to compute fire-averaged EFs in fresh smoke from landscape fires. Then we address the accuracy of mass-concentration measurements for PM$_{2.5}$ and eBC from the AM520 and the AE51, respectively.



### 3.1 Emission factor measurements

During 24 prescribed experimental fires in the KNP, we measured mixing ratios of CO, $CO_2$, $CH_4$, $N_2O$, $PM_{2.5}$, and eBC at the top of a 15-meter mast. Figure 1 is an example of a temporal concentration profile from a prescribed fire experiment at the Skukuza EBPs in August 2017. The red horizontal line represents the mixing ratio of and the sample time in the bag sample.

$CO_2$ concentration enhancements dominated the passing of the fire front but diminished as RSC took over. After the flaming phase ceased, mixing ratios and thus temporal varying EFs (green lines) for CO, $CH_4$ and $PM_{2.5}$ sharply rose. Emissions for these species persisted for the entire duration of the measurement.

Comparing EFs based on the integrated mast measurements with averages of UAS-filled bags indicated a good agreement with
$R^2$-values ranging from 0.81 to 0.95. Fig. 2 represents the EF, calculated from the cumulative emissions of the UAS-sampled bags, plotted against EFs calculated from cumulative emissions that passed the mast with each point representing a single fire (11 fires in total).

### 3.2 PM mass concentration in BB smoke

To determine the AM520 CF for BB particles, we compared the PM mass concentrations measured by the AM520 to the mass
concentration derived from the particle size distribution measured by the SMPS (Fig. 3a), and the gravimetrically using TEOM and filters (Fig. 3b). Average PM mass concentrations during the TROPOS experiments, derived from the SMPS, were 0.35 mg m$^{-3}$ for hay, 0.14 mg m$^{-3}$ for wood and 0.08 mg m$^{-3}$ for wood chips emissions. Fuel-specific AM520 CFs calculated using the SMPS as a reference, were 0.23 for hay, 0.26 for wood, 0.29 and for wood chips emissions. Using an averaged CF of 0.27, linear correlation of $PM_1$ mass concentrations had an $R^2$ of 0.85., the average CF for peat fires calculated using the TEOM as
a reference for five AM520's was 0.17

Particles were small with volumetric median particle diameters (VMDs) of 183, 162 and 184 nm respectively for wood chip-, wood- and hay fires. We did not find significant correlations of the CF with either the VMD, the eBC- and BrC concentrations or the absolute $PM_{2.5}$ mass concentration measured by either instrument. During the chamber experiments at the KWTC, cross-
correlation of the AM520 with five co-located AM520 modules revealed deviations ranging from -20% to +12%. The relative errors for the respective AM520 reference modules were constant, and could therefore be corrected for by applying unit-specific CFs for the different AM520s.

### 3.2 Black carbon mass concentration in BB smoke

Black carbon was measured using the three absorption-based measurement techniques (eBC) described under *Methods* as well
as through thermal-optical analysis of filter samples (EC). During the experiments at Tropos, the average eBC concentrations





measured by AE33 at 880 nm were 30.92 μg m⁻³ for hay, 19.64 μg m⁻³ for wood and 18.65 μg m⁻³ for wood chips emissions. We found a strong agreement ($R^2$ = 0.93) for the Weingartner-corrected eBC ($\lambda$ = 880 nm) measured by the AE51 and AE33 aethalometers (Fig. 4a). However, at low concentrations, AE33 measurements were 30-70% higher than AE51 measurements. While closer to unity, linear correlation of the wavelength-adjusted absorption coefficient to the MAAP absorption coefficient

was less robust with an $R^2$ value of 0.77 (Fig. 4b).

To assess the importance of BrC absorption and wether its effect can be neglected at a wavelentght of 880 nm, we calculated $\text{Å}_{ap}$ for the total fit of the AE33 wavelength and the separate BC- and BrC fractions, in the TROPOS experiments. At wavelengths over 750 nm, absorption almost completely follows the BC-curve (Fig. 5) indicating that contribution of BrC

absorption was small (difference in absorption of <10%), whereas absorption in the ultra-violet was dominated by BrC. The absorptive Ångström exponents ranging from 1.2 to 5.5 indicated high BrC concentrations. This indicates that in the case of BB, assuming the absorption at 880 nm is solely due BC slightly overestimates the BC concentration.

Figure 6 shows the empirically-derived MAC values for the different experiments. These MAC values, derived from the

relation between the AE51 absorption coefficient and the Sunset EC mass concentration, were highly variable, ranging from 1−17 m² g⁻¹ with an average of 5.56 ± 5.05 m² g⁻¹. BB studies suggest a MAC of 4.7 m² g⁻¹ at 880nm for fresh uncoated BC (Bond and Bergstrom, 2006; Kumar et al., 2018). In comparison, the static MAC-value assumed by the AE51 is 7.8 m² g⁻¹. If no fire-specific MAC value can be determined, we propose a correction factor (CF$_{MAC}$) of 0.72 to compensate for the difference between the MAC value assumed by the firmware (7.8 m² g⁻¹) and the empirically-derived MAC value for fresh BB particles

(5.6 m² g⁻¹). Note that the axes in Fig. 8 are on a logarithmic scale and the average MAC-value from the landscape fires in KNP was more than double what we found in the laboratory measurements. Moreover, MAC-values for individual KNP fires ranged from 3.3 to 16.8 m² g⁻¹.

We used the empirically determined BB correction factors from the laboratory fires (AM520 CF = 0.27 and MAC CF = 0.72)

to calculate aerosol EFs from the KNP fires. Fig. 7 presents the EF's for PM$_{2.5}$ and BC plotted against MCE. Since BC, PM$_{2.5}$ and carbonaceous trace gasses were measured at 1-second frequency, we can calculate the EF for every second of the mast measurements (small dots) as well as the fire averages (crosses). While we found a clear negative correlation of the PM$_{2.5}$ EF with MCE, our results did not indicate a significant MCE correlation with the eBC EF. The PM$_{2.5}$/MCE regression line crossed 0 when MCE reached unity, whereas BC measurements from the aethalometer were still significant. This meant that during

high-MCE combustion, the EF for OC diminished causing the BC contribution to PM$_{2.5}$ to increase exponentially.



## 4. Discussion

Given that the comparison of UAS-based and mast-based measurement was encouraging and straightforward, we focus the discussion on the implications for the calculated EFs in the KNP. Then we address the performance of the individual measurements and the empirically derived correction factors for BB particles. Finally, we elaborate on the uncertainties
associated with these measurements.

### 4.1 Field derived emission factors

Due to the large spatio-temporal variability in vegetation and weather conditions and the unpredictable nature of landscape fires, comparing and extrapolating BB EFs is challenging. Using the empirically derived correction factors from the laboratory BB experiments (AM520 CF = 0.27, MAC CF = 0.72), EFs from our KNP measurements were in line with previous savanna
burning studies, albeit that MCE was relatively high compared to earlier measurements using FTIR (Table 3, Andreae., 2019; Wooster et al., 2011). For aerosol emissions, the literature studies listed by Andreae (2019) include a variety of different methods; $PM_{2.5}$ measurements were performed using OPCs (McMeeking et al., 2006), nephelometer (Burling et al., 2011; Cachier et al., 1995; McMeeking et al., 2006), SMPS (Desservettaz et al., 2017) or gravimetric filter analysis (e.g. Alves et al., 2010; Cachier et al., 1995; Korontzi et al., 2003; Ward and Radke, 1993; Yokelson et al., 2013), whereas BC and EC
measurements studies were performed using Thermal Optical Reflectance (TOR) (Alves et al., 2010; McMeeking et al., 2006; Yokelson et al., 2013), Aethalometer measurements (McMeeking et al., 2006) or coulometric titration (Cachier et al., 1995).

The $PM_{2.5}$ EF showed a clear MCE dependence which corresponded with previous literature findings (Collier et al., 2016; Yokelson and Ward, 1996). Contrary to the total $PM_{2.5}$ EF, we found significant BC emissions, even when the MCE
approached unity. Liu et al. (2014) and Pokhrel et al. (2016) found a similar exponential relation for the BC:TC ratio with the MCE for both laboratory and landscape fires. This resulted from a diminishing OC EF, rather than an increase of the BC EF. While we did not find a significant correlation between BC EF and MCE, the fuel type appeared to be significant for the BC EF since grass-dominated Satara plots emitted up to 3 times more BC -per unit of fuel at the same fire-average MCE- than more tree-covered Pretoriuskop plots (Table 3).

The high contribution of BC to the emitted $PM_{2.5}$ at high-MCE combustion indicates a lower single scattering albedo (SSA) (Brown et al., 2021; Liu et al., 2014; Pokhrel et al., 2016). In savannas, the MCE is thought to increase over the course of the dry-season, as the vegetation senesces and humidity drops (Korontzi et al., 2003; Vernooij et al., 2021). Eck et al. (2013) studied seasonal changes of BB particles during 15 annual fire seasons in southern Africa, using the Aerosol Robotic Network
(AERONET). Contrary to the expectation that seasonal increase in MCE should lead to a decrease of the SSA, they found a linear trend of SSA increasing throughout the dry season. More direct measurements of seasonality in BB aerosol properties may clarify the effects of both human- and climate-induced fire regime shifts on radiative forcing.



## 4.2 Performance of the individual measurements

### 4.2.1 Gas measurements

The stability of GHG samples in Tedlar® bags has been previously demonstrated (Meyer et al., 2012; Silva et al., 2016) and cavity ring-down spectroscopy has been shown to be a stable and accurate method for GHG measurements under both laboratory- and field conditions (Yver Kwok et al., 2015). We frequently calibrated the CRDS analysers, and Tedlar® bags with calibration gas were measured interspersed with the smoke samples to determine the CRDS stability. Field measurements of 35 sample bags with calibration gas, spread out over 4 measurement campaigns, showed an average underestimation of -4.57% for $CO_2$, -1.73% for CO, -3.59% for $CH_4$ and -1.36% for $N_2O$ compared to the known reference gas composition. As calibration schemes for both analysers are linear, these underestimations were linearly transferred to the sample measurements.

### 4.2.2 eBC mass concentration measurements

We found that the lightweight AE51 aethalometer agreed well with both the MAAP and the AE33 multiwavelength aethalometer. This was consistent with previous measurements for city pollution (Alas et al., 2019; Pikridas et al., 2019), personal BC exposure (Cai et al., 2013), vertical atmospheric profiles (Ferrero et al., 2011), seasonal background fluctuations (Zhao et al., 2019) and crop burning emissions (Zhang et al., 2015). Pikridas et al. (2019) tested the use of UAS-fixed aerosol absorption sensors including the AE51 in ambient and diluted city pollution and found a similar correlation with BC measured by the MAAP with an $R^2 = 0.76$ for a slope of 0.94. We did not find the same relation between the MAAP and the AE51 eBC measurements found by Alas et al. (2019) and Pikridas et al. (2019) for city pollution. A possible explanation for this is that biomass combustion is typically associated with higher emission levels of BrC compared to other BC sources, which would disproportionately affect the MAAP and AE51 measurements. The MAAP operates at a lower wavelength (631 nm) than the AE51 (880 nm) meaning that the absorption coefficient of the MAAP is more sensitive to BrC. We found average absorptive Ångström exponents of 4.55, 4.67 and 5.55 for the BrC fraction from wood chips, hay and wood combustion emissions, respectively. The measured absorptive Ångström exponents were high compared to field measurements for BB smoke in Africa and Brazil that ranged from 0.8 to 2.1 (Reid et al., 2005b), but in line with BrC absorptive Ångström exponents from BB studies listed in Pokhrel et al. (2016), ranging from 3 to 19.

Although our results show a strong correlation between the AE51 and the AE33 aethalometer at the same wavelength, the AE51 underestimated eBC at low concentrations. At mass concentrations over 50 μg m⁻³, accuracy improved. The lower accuracy at low concentrations may in part be related to the reduced sensitivity caused by the low flow-rate at which we operate the AE51 rather than C-value. Nonetheless, the agreement between both aethalometers, considering corrections for wavelength, temporal resolution and sensitivity was robust. Our measurements suggest C-values in the 2.14–2.78 range. In comparison: Ferrero et al. (2011) found an optimal C-value of 2.05 ± 0.03, whereas C-values found by Weingartner et al.





(2003) were in the 2.13–3.90 range.

There is no 'gold standard' for measuring BC concentrations, thus the use of different wavelengths, filter material, illumination angles, etc. makes comparing methods challenging. Discrepancies in filter changes, filter loading effect and differences in temporal resolution could cause some of the variability found in the MAAP and AE51 measurements. Although the MAAP and AE33 aethalometer are often used as reference equipment, both are filter-based methods that assume a fixed MAC just like the AE51 and are therefore sensitive to scattering (Müller et al., 2011).

The ambiguous use of terms like soot, EC, and eBC is often problematic when comparing studies. Although eBC and EC measure different properties, they are thought to be largely overlapping and EF compilations like Akagi et al. (2011) and Andreae (2019) therefore list them in the same column without conversion. Moreover, EC measurements are used to calibrate the MAC value that aethalometers use to derive BC mass from the absorption coefficient. Even though BC is equated to EC for the purpose of calibration, EC to BC ratio measurements in BB literature are highly variable ranging from 0.3 to 1.6 and appear to be strongly related to the aerosol type, the degree of atmospheric processing (Rigler et al., 2019; Salako et al., 2012) and the MCE of the fire (Aurell et al., 2017). Using the manufacturer-defined MAC of 7.8 $m^2$ $g^{-1}$, BC concentrations and the difference compared to the EC concentrations were 223 $\mu g$ $m^{-3}$ (-71%), 16 $\mu g$ $m^{-3}$ (-56%) and 57 $\mu g$ $m^{-3}$ (+20%), for the experiments at FLARE, TROPOS and KNP, respectively. Even if we apply our empirically-derived MAC of 5.2 $m^2$ $g^{-1}$ for all measurements, BC concentrations and difference with the EC concentrations were still 335 $\mu g$ $m^{-3}$ (-56%), 24 $\mu g$ $m^{-3}$ (-34%) and 82 $\mu g$ $m^{-3}$ (+76%). The large uncertainty in the MAC value is possibly related to the fact that EC and eBC measure different properties and are susceptible to different types of measurement errors (Schmid et al., 2001). While the AE51 agrees well with high-fidelity filter-based methods, the high variability in MAC values in BB smoke remains a weak-point for BC mass derivation from absorption-based measurements.

### 4.2.3 PM$_{2.5}$ mass concentration

We found an overall AM520 CF of 0.27 for an optimal correlation of PM$_1$ concentrations from the AM520 and the SMPS derived PM$_{1.0}$ with an $R^2$ of 0.85. The AM520 CF ranges for individual fuel types were 0.14–0.42 for wood, 0.22–0.24 for hay and 0.27–0.31 for wood chips. These CFs are low compared to those found by previous BB studies (Table 4).

Our CF was closer to Stauffer et al. (2020), who calibrated the AM520 against a beta ray attenuation monitor for diluted wildfire smoke and found a CF of 0.14. Using the gravimetric TEOM and filter measurements in PM$_{2.5}$, from peat smoke, we found an average AM520 CF of 0.17 with a range of 0.07–0.32 - somewhat lower than previous field measurements in tropical peat fires by Wooster et al. (2018) using the TSI Dusttrack. This may indicate substantial differences in the laboratory compared to field analyses for PM$_{2.5}$ mass concentration. Cross-correlation of the UAV-mounted AM520 to five co-located





AM520 modules revealed measurement errors of up to 20%. The error margins between individual units were constant and could be compensated for by unit-specific CFs. While the UAV-mounted unit was freshly factory-calibrated, the reference units were not, after a year of intensive use. This could be a potential explanation for drift which could be remedied by recalibrating.

Like all OPCs, the AM520 computes particle mass concentration from particle counts based on the scattering of light by individual particles. The mass scattering efficiency (MSE) is dependent on particle size and the refractive index of the particles. For particles in the Rayleigh regime with a size parameter (the ratio of particle circumference $\pi D$ to wavelength of the light $\lambda$) smaller than unity, MCE is proportional to the particle diameter cubed, whereas for larger particles MSE becomes inversely

proportional to particle diameter (Moosmüller and Sorensen, 2018b). At the measurement wavelength of the AM520, a size parameter of 1 corresponds to a particle size of approximately 200 nm. Fig. 10a and 10b show that a large portion of the BB particles measured in our laboratory experiments falls within the Rayleigh scattering regime. This indicates scattering by BB particles in the AM520 is strongly size-dependent. As the AM520 does not measure different size bins, it is not possible to use a size-resolved MSE to compute the mass concentration. The median diameter of Arizona roadside dust used for the AM520

factory calibration is 2.12 µm. Therefore, Mie scattering rather than Rayleigh scattering is the dominant scattering regime for these particles. Particle MSE at this diameter corresponds to roughly 1 $m^2\,g^{-1}$ while at the VMD of the measured BB particles, MSE is likely to be higher (Moosmüller and Sorensen, 2018a; Rogers et al., 2005).

We did not observe a significant correlation between the AM520 CF and VMD or BC to PM ratio, albeit that the overall

difference in particle size distribution between experiments was low (Fig. 8). With respective VMDs for particles from wood chips, wood, and hay of 183, 162 and 184 nm, the average particle size was small compared to savanna and grassland fire studies in fresh smoke with VMD ranging from 230−300 nm (May et al., 2014; McClure et al., 2020; Reid et al., 2005a). Wildfires typically burn less efficient compared to controlled BB under laboratory conditions (Park et al., 2013). In contrast to our findings, Dacunto et al. (2013) found the AM520 CF to be dependent on the VMD of the emitted particles, indicating

that the lab-derived CFs may not be representative for the field measurements.

### 4.3 Assumptions and uncertainty analysis

There are various caveats associated with the methodology that are important for our results and further application. Here we will discuss the nature of the uncertainties and how we address them in our measurements.

### 4.3.1 Variability in the mass absorption cross-section for airborne BC particles

For BC measurements, the largest uncertainty originates from the mass absorption cross-section for airborne BC particles ($MAC_{air}$) which is used to convert absorption of airborne particles to mass concentration. In general, the $MAC_{filter}$ (used to


convert light attenuation by a loaded filter) equals the $MAC_{air}$ multiplied by the empirically determined multiple scattering parameter (C) (Liousse et al., 1993; Pikridas et al., 2019; Weingartner et al., 2003). The C-value typically ranges from 1.5-2.5 and depends on the particle type as well as the filter material used and is therefore instrument dependent.

For the conversion of attenuation of light to airborne BC mass concentrations, the standard $MAC_{air}$ values used by our analysers are 6833 / λ $m^2$ $g^{-1}$ (7.8 $m^2$ $g^{-1}$ at 880 nm) for the aethalometers (Drinovec et al., 2015) and 6.6 $m^2$ $g^{-1}$ at 637 nm (which corresponds to 4.7 $m^2$ $g^{-1}$ at 880nm) for the MAAP (Müller et al., 2011). Although the instruments assume fixed MAC values, the MAC in itself is dependent on the BC to OC mixing ratio (Cappa et al., 2020; Sandradewi et al., 2008), the particle size distribution, the structure of the measured BC (Conrad and Johnson, 2019; Petzold et al., 1997; Zhao et al., 2019) and the

coating of soot particles by organic particles (Adachi et al., 2010; Cappa et al., 2020). MAC has an inverse linear wavelength-dependency (Zhao et al., 2019):

$$MAC(\lambda) = MAC(\lambda_0) \times \left(\frac{\lambda}{\lambda_0}\right)^{-Å_{ap}} \tag{15}$$

For BC within the Rayleigh regime, $Å_{ap}$ is close to unity (Conrad and Johnson, 2019). If we assume BC in fresh smoke to exist as uncoated particles, a $MAC_{air}$ of 7.5 $m^2$ $g^{-1}$ at a wavelength of 550 nm is advised (Bond and Bergstrom, 2006; Cheng et al., 2016). Following Eq. (15) this translates to a $MAC_{air}$ of 4.7 $m^2$ $g^{-1}$ at a wavelength of 880 nm which is consistent with the MAC found by Kumar et al. (2018) and the MAC used by the MAAP. However, rapid coating of BC with non-absorbing liquid organic compounds once emitted may lead to much higher values. Liousse et al. (1993, 1995) found a high $MAC_{air}$ of

$15 \pm 5$ $m^2$ $g^{-1}$ for BB smoke from savannas. Aircraft measurements from Brazilian wildfires during the SCAR-B campaign also found high $MAC_{air}$-values ranging from 5.2 to 19.3 $m^2$ $g^{-1}$ with an average value of 12.1 $m^2$ $g^{-1}$ (Martins et al., 1998) at 550 nm. This may in part explain why we found MAC values for laboratory studies to be much lower than those measured in KNP landscape fires.

At the AE51 measurement wavelength of 880 nm, the MAC is thought to be relatively stable (Cappa et al., 2020), whereas at shorter wavelengths BrC causes larger fluctuations. However, also at the 880 nm wavelength we found highly variable MAC values for different fires with an average of $5.2 \pm 5.1$ $m^2$ $g^{-1}$. This confirms earlier findings by Salako et al. (2012), that universally applying MAC values to aerosols with different optical properties thus possibly results in large measurement errors. On the UAS, we deployed co-located EC measurements allowing for fire-averaged MAC values. We found that accounting

for variability within individual fires was difficult as separate filters for smaller peridos of the fire (e.g. the flaming and smouldering phase) resulted in insufficient filter loading.





### 4.3.2 Filter loading effect

The filter loading effect is considered one of the major weaknesses of the AE51 measurement (Drinovec et al., 2015; Good et al., 2017; Weingartner et al., 2003). Although loading correction is not required below a threshold value of 10−20% (Weingartner et al., 2003), we corrected all AE51 measurements using the Weingartner correction (Sect 2.3.2). For the
observed attenuation range of 0−35% (compared to the initial laser transmittance), we found a slight average decrease in Weingartner-corrected AE51 eBC compared to the AE33 eBC measurements of 0.21% per % of attenuation (Fig. 9). This is consistent with findings by Good et al. (2017) who noted that the Weingartner correction tends to undercompensate while the method used by the AE33 tends to overcompensate at this interval. In the field, filters are changed every fire and the flow rate is set to 50 mL min⁻¹. Therefore, although we expect high BC concentrations, we expect performance loss due to filter loading
to be limited. However, if attenuation is higher than 35%, we may revert to other load-compensation methods described by Good et al. (2017).

### 4.3.3 Relative humidity and temperature

At high environmental humidity levels, density, refractive index and morphology of aerosol particles change as a result of $H_2O$-condensation. If not accounted for, hygroscopic growth may affect the MSE and therefore reduce measurement accuracy
of OPCs (Gu et al., 2016; Jayaratne et al., 2018; Li et al., 2018; Mehadi et al., 2020). For relative humidity (RH) between 60% and 95%, Gu et al. (2016) found an exponential increase in the diameter of spherical particles leading to a linear decrease in the refractive index. Aethalometer measurements are also sensitive to sudden changes in RH and temperature (Cai et al., 2013; Nessler et al., 2006). Water entering the filter may cause the fibres in the filter to swell causing an increase in light scattering. To our knowledge, RH is not adjusted for by either the AM520 or the AE51. Adding a diffusion dryer can eliminate these
problems, however it may also affect measurements due to losses and discrepancies between the measurement and reference conditions.

Hygroscopic growth occurs when the relative humidity exceeds the deliquescence point of a chemical substance (Jayaratne et al., 2018). Semeniuk et al. (2007) studied the hygroscopic behavior of BB aerosols under an environmental transmission
electron microscope. They found that ambient particles from biomass burning smoke had a relatively high deliquescence point and typically took up water in the range 80–100% RH. Our own unpublished RH measurements, covering over 2400 UAS bag sample EF measurements in dry-season savannas show a RH-range of 1.5% to 47.3% with an average of 17% and a standard deviation of 6.8%. The average variation within a single fire was 10%, which resulted from the diurnal temperature cycle rather than the sudden change that would affect BC measurements. The high BC concentrations found in BB plumes allows
for a very low flowrate (50 mL min⁻¹) and we did not find the distinctive negative spikes in BC associated with humidity drops. Therefore, we do not expect significant condensation-induced effects on either the PM or the BC measurement.





### 4.3.4 Effect of BC on PM measurement performance

The large differences in particle characteristics from smouldering and flaming combustion products results in an inhomogeneous composition of BB aerosols (Moosmüller et al., 2009). One of the greatest uncertainties in OPCs originates from the poorly understood, complex refraction index of soot particles (Sorensen, 2001). Increased concentrations of BC and

BrC lead to a decrease in scattering efficiency of the total aerosol mass. OPCs that rely on fixed scattering indices for the conversion to aerosol mass may underestimate the total mass concentration due to reduced scattering in BB particles. Also, BC particles tend to have a non-spherical morphology (Chakrabarty et al., 2006). The assumption of sphericity made by OPCs is therefore incorrect and light scattering becomes much more complex (Sorensen, 2001). Mehadi et al. (2020) found that for several low-cost OPCs, the EC/OC ratio had a significant impact on the measurement accuracy, with higher ratios leading to

lower OPC readings compared to a BAM 1020 reference instrument. BC exists as agglomerates which are built up of spherules with individual diameters as small as 30 to 50 nm (Chakrabarty et al., 2006). Combined with their low SSA of 0.46 (Müller et al., 2011), this may lead to insufficient scattering by some BC particles to be detected by the AM520. This may explain why the $PM_{2.5}$/MCE regression line crossed 0 when MCE reached unity, whereas BC measurements from the aethalometer were still significant. Since the SMPS uses impaction rather than light scattering to count aerosols, SMPS measurements should not

be affected by optical properties of particles. We did not find significant dependence of the AM520 CF on the eBC to $PM_1$ ratio.

### 4.3.5 Calculation through density

Carbonaceous particles, in particular BC, have lower density compared to mineral dust. Zhai et al. (2017) calculated the effective density of BB particles in the particles in the size range of 50–400 nm. They used an aerosol particle mass analyser

to measure the mass of particles that had been classified according to electrical mobility by a differential mobility analyser DMA. They found dominant density modes in the effective density distributions of 200 and 400 nm mobility-selected particles of 1.40 and 1.35 g cm$^{-3}$ respectively. In this study, particle density for the SMPS was assumed to be constant at 1.5 g cm$^{-3}$, which is a typical density for wood burning primary organic aerosols based on Kumar et al. (2018) and Moosmüller et al. (2009).

### 4.4 Caveats of a UAS-based approach

According to Ward and Radke (1993), to evaluate an "average" EF or emission ratio that is representative of the overall flaming and smouldering combustion phases, the emissions must be sampled at a rate proportional to the rate of carbon release in each phase over the duration of the fire. This was done by Wooster et al. (2011) using airborne fire radiative power (FRP) measurements made concurrent with the trace gas observations over the Kruger park fires. Such FRP measurements were not

available where, and although we measured the atmospheric concentrations continuously as the different stages of combustion





products passed the mast, we did not measure the fluxes. During the flaming phase, updraft of the hot reaction products is much more rapid than during the residual smouldering phase (Ward and Radke, 1993). We let the mast measurement run for as long as possible but were limited by analyser battery capacity. In some experiments, including the example in Fig. 1, small peaks of RSC-emissions like $CH_4$, CO and $PM_{2.5}$ were still recorded upon shutting down the measurement. This indicates a

slight underestimation of the significance of RSC for these plots. Nonetheless, the low contribution of RSC was consistent with previous mast measurements in Brazilian savanna from Ward et al. (1992) as well as previous studies in the KNP (Cofer et al., 1996; Wooster et al., 2011). In savanna vegetation, grass and fine fuels dominate the fuel mixture and the contribution of RSC-prone fuels is limited. Vegetation types where the portion of fuel combusted in RSC is more substantial (e.g. forests and peatlands) may call for a different measurement approach.

While the proposed UAS-based sampling method lacks the high temporal resolution of continuous EF measurement from the mast, the ability to follow the fire front as it passes through the landscape makes it much easier to obtain large amounts of measurements. Erecting a mast is tedious and time-consuming. Many attempts led to non-ideal measurements, for example when the wind direction changed and blew the smoke away from the top of the mast, or when the fire front did not spread and

burn the vegetation surrounding the mast, or the 'backfire' did. UAS measurements like those described by Vernooij et al. (2021) are more versatile, e.g. allowing 60−80 gas samples for a single fire, over the course of several hours. This results in a much higher coverage of the spatial variability in the fuel, and the temporal variability in fire characteristics as weather conditions change.

### 5. Conclusion

We propose a UAS-based methodology for measuring GHGs and aerosol EFs in fresh smoke from landscape fires. In a series of laboratory and field experiments we addressed the main uncertainties considered with the methodology and calculated correction factors for the measurement of fresh BB particles. We tested our UAS setup against a continuous measuring mast and calculated fire-averaged emission factors using both setups. Overall, fire-averaged emission factors from the UAS agreed well with measurements from the mast and were in line with previous BB literature. While variability in the $PM_{2.5}$ EF was

well-explained by the MCE, we found no significant correlation between MCE and BC EF. UAS allows for flexibility as fire behaviour and weather conditions change as well as sample at different heights within the plume.

Although our results highlight the potential of UAS-based EF measurements, aerosol EFs measurements remain prone to several uncertainties related to atmospheric processes. Particle mass concentration measured by the lightweight AM520 was

compared to an SMPS and gravimetric filter measurements. We found optimal calibration factors to use for the AM520 in BB smoke to be 0.27. However, with calibration factor ranges of 0.14−0.42 for wood, 0.22−0.24 for hay, 0.27−0.31 for wood chips and 0.07–0.32 for peat samples, there was significant variability between fires. Equivalent black carbon (eBC) mass

concentrations from the AE51 aethalometer agreed well with eBC measurements from the AE33 at a wavelength of 880 nm and absorption coefficient measurements from the MAAP. Optimal agreement was achieved using a correction factor of 1.3, though this may be related to low overall concentrations (<50 µg m$^{-3}$ eBC). A caveat for eBC measurements, indicated by both our own findings and previous literature, is that a pre-set manufacturer MAC cannot be universally applied to BB

measurements. BB particle properties and atmospheric conditions are highly variable, which resulted in a wide range of MAC-values (2.1−25.4 m$^2$ g$^{-1}$) for the individual fires we measured. This indicates that MAC correction with EC remains a continuous necessity, to reduce this uncertainty. This is not unique to light-weight aethalometers but affects all methods that use fixed MAC values to calculate eBC mass concentrations from absorption coefficients. We found that in fresh smoke, the contribution of BrC to the total absorption of BB particles was significant. Measurements at an additional short-wavelength

band may therefore benefit absorption measurements. While significant uncertainty remains for both the eBC from the AE51 aethalometer and PM$_{2.5}$ from the AM520 optical particle counter, much of this uncertainty is inherently associated with aerosol mass-derivations from optical properties, and thus similarly applicable to high-fidelity analysers. Overall, we found that quality of the data is sufficient to measure EFs in fresh biomass smoke if proper corrections are applied, and the described caveats are avoided.

**Author's contributions:**

RV, GRvdW, PW and NS designed the study; RV, PW, MW, LP and MG conducted the experiments in the laboratory; RV, PW, MW, GR and TS conducted the field measurements; RV and UD conducted the aerosol analyses on the samples. RV wrote the manuscript with help from PW, MW, NS and GRvdW.

**Competing interests:**

The authors declare no competing interests.

**Data Availability:**

All the data presented in this study are available from the authors upon request.

**Acknowledgments**

This research has been supported by the Netherlands organization for Scientific Research (NWO) through the Vici scheme

research programme, no. 016.160.324. This work has also received funding from the European Union's Horizon 2020 research and innovation program through the EUROCHAMP-2020 Infrastructure Activity under grant agreement No 730997. Part of



this work was supported by the COST Action CA16109 COLOSSAL Chemical On-Line composition and source apportionment of fine aerosol.

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



**Table 1: Description of the dominant vegetation and relevant characteristics for the experimental sites.**

| Landscape ID | Dominant vegetation[1] | Mean annual rainfall (mm)[1] | Average fire freq.[2] | Measured fires |
|---|---|---|---|---|
| Skukuza | Savanna dominated by dense *Combretum collinum*/ *Combretum zeyheri* trees | 550 | 3.63 years | Apr: 4 Aug: 1 Oct: 5 |
| Satara | Open grassland savanna with scattered tall (10–15 m) *Marula* (*Sclerocarya birrea*) and knobthorn Acacia (*Acacia nigrescens*) trees | 544 | 4.33 years | Apr: - Aug: 6 Oct: - |
| Pretoriuskop | Savanna dominated by dense tall (10–15 m) clusterleaf (*Terminalia sericea*) trees | 737 | 2.22 years | Apr: 2 Aug: 2 Oct: 1 |
| Mopane | Savanna dominated by dense low (1–2 m) mopane (*Colophospermum mopane*) trees | 496 | 4.57 years | Apr: 2 Aug: 1 Oct: - |

[1] based on Table 1 of Govender et al. (2006).
[2] based on the average Landsat derived continuous fields of tree cover 2015 (Sexton et al., 2013) and MODIS-based vegetation continuous fields dataset (MCD44Bv6, Townshend et al., 2011) in the mapped area of the vegetation types (Gertenbach, 1983).



**Table 2: Description of analysis equipment used for UAS measurements of EFs in biomass burning plumes**

| Analyzer | Measurement technique | Measured species | Measurement precision | Calibration |
|---|---|---|---|---|
| Los Gatos micro-portable $CO_2/CH_4$ analyzer | Off-axis integrated-cavity output spectroscopy | $CO_2$<br>$CH_4$ | 2 ppm<br>3 ppb | *Standard gas conc.*<br>4968 ppm ($\pm$ 2%)[1]<br>15.71 ppm ($\pm$ 5%)[1] |
| Aeris Pico mid-IR Laser-based $CO/N_2O$ analyzer | Cavity ringdown spectroscopy | CO<br>$N_2O$ | 1 ppb<br>1 ppb | *Standard gas conc.*<br>103.0 ppm ($\pm$ 2%)[1]<br>1.15 ppm ($\pm$ 2%)[1] |
| TSI SidePak® AM520 Optical particle counter | Optical particle counter 90º light scattering of 650 nm laser diode | Particulate matter < 2.5 µm | 1 µg m$^{-3}$<br>Range: 0.001-100 mg m$^{-3}$ | *Calibrated against:*<br>1. TROPOS SMPS[2]<br>2. Tapered Element Oscillating Microbalance (gravimetric)[3]<br>3. cross-calibration (5x AM520)[3] |
| MicroAeth® AE51 black carbon analyzer | Attenuation of $\lambda$ = 880 nm light by a particle-loaden filter $MAC_{air}$ = 7.77 | Equivalent Black Carbon | 0.1 µg m$^{-3}$ | *Calibrated against:*<br>1. Aethalometer AE33 ($\lambda$ = 880)[2]<br>2. MAAP 5012 ($\lambda$ = 637)[2]<br>3. Sunset analyser (thermo-optical EC)[4]<br>4. cross-calibration[3] |

[1] Fire Laboratory of Amsterdam for Research in Ecology (FLARE), Amsterdam
[2] Leibniz Institute for tropospheric research (TROPOS), Leipzig
5  [3] Kings Wildfire Testing Chamber (KWTC), London
[4] Centre for Isotope Research (CIO), Groningen



**Table 3: Emission factors (g kg$^{-1}$) measured using the UAS-method as well as those listed in Andreae (2019)**

| | Satara 6 EBPs | Skukuza 4 EBPs | Mopani 1 EBP | Pretoriuskop 2 EBPs | KNP avg. FTIR 4 EBPs (Wooster et al, 2011) | Savanna avg. (Andreae, 2019) |
|---|---|---|---|---|---|---|
| FTC (LS)[1] | 4.96% | 5.56% | 5.23% | 8.60% | - | - |
| $CO_2$ EF | 1668 | 1643 | 1656 | 1607 | $1665 \pm 54$ | $1660 \pm 90$ |
| CO EF | 44.9 | 52.8 | 50.8 | 66.6 | $101 \pm 30$ | $69 \pm 20$ |
| $CH_4$ EF | 1.0 | 1.9 | 1.2 | 3.0 | $2.5 \pm 0.9$ | $2.7 \pm 2.2$ |
| $N_2O$ EF | 0.25 | 0.29 | 0.47 | 0.15 | - | $0.17 \pm 0.09$ |
| $PM_{2.5}$ EF | 4.8 | 9.2 | 3.5 | 6.2 | - | $3.6 \pm 0.02$ |
| eBC EF | 0.99 | 0.73 | 0.68 | 0.50 | - | $0.53 \pm 0.35$ |
| MCE | 0.95 | 0.93 | 0.96 | 0.94 | $0.91 \pm 0.06$ | $0.94 \pm 0.02$ |

[1] Landsat derived continuous fields of tree cover 2015 (Sexton et al., 2013)

**Table 4: AM520 CF results compared to previous studies**

| Fuel | CF | Reference | Study |
|---|---|---|---|
| Logs | 0.14 – 0.42 | SMPS | This study |
| hay | 0.22 – 0.24 | SMPS | This study |
| wood chips | 0.27 – 0.31 | SMPS | This study |
| Peat | 0.07 – 0.32 | TEOM, gravimetric filter | This study |
| cherry wood logs (fireplace) | $0.44 \pm 0.01$ | Gravimetric filter | Dacunto et al., (2013) |
| Wood chips | $0.77 \pm 0.07$ | Gravimetric filter | Jiang et al. (2011) |
| Wildfire smoke (dilute) | 0.14 | beta ray attenuation monitor | Stauffer et al., (2020) |
| Peat fires | $0.5 \pm 0.09$[1] | Gravimetric filter | Wooster et al., (2018) |
| Forest fire | $0.45 – 0.7$[1] | beta ray attenuation monitor/ Gravimetric filter | McNamara et al., (2011) |
| wood- and coal-smoke | $0.37$[1] | TEAM, high volume gravimetric filter | Kingham et al., (2006) |

5  [1] Calibration factor determined for DustTrak™ (Same initial reference: ISO 12103-1 A1)



**Table A1: Abbreviations and their definitions**

| | |
|---|---|
| $(e)BC$ | (Equivalent) Black Carbon |
| EC | Elemental carbon |
| $\sigma_{ATN}$ | Attenuation coefficient |
| $\sigma_{Ap}$ | Absorption coefficient |
| C | Optical enhancement factor |
| $\omega_0$ | Single Scattering albedo |
| $MAC_{air}$ | Mass absorption coefficient of particles suspended in air |
| $MAC_{filter}$ | Mass Absorption Coefficient of particles on loaded on a filter |
| $\text{Å}_{ap}$ | Ångström exponent |
| $EF_i$ | Emission factor of species (i) |





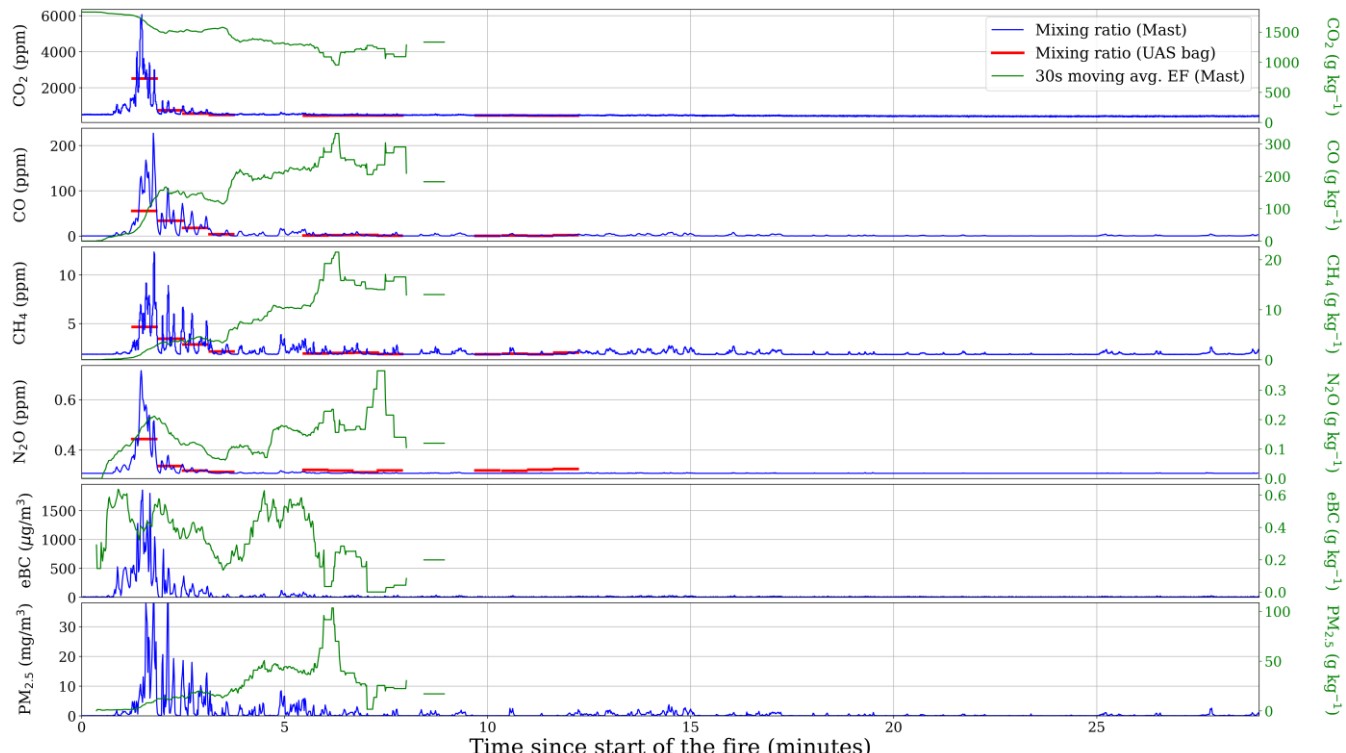

**Figure 1: Example of a mast measurement profile (blue, left y-axis represents the absolute concentration) for a prescribed burn near Skukuza at the KNP in August and the corresponding emission factors (green, right y-axis). The red stripes represent the timing and measurement of the UAS sampled bags. The UAS was positioned close to the measurement inlet on the mast.**

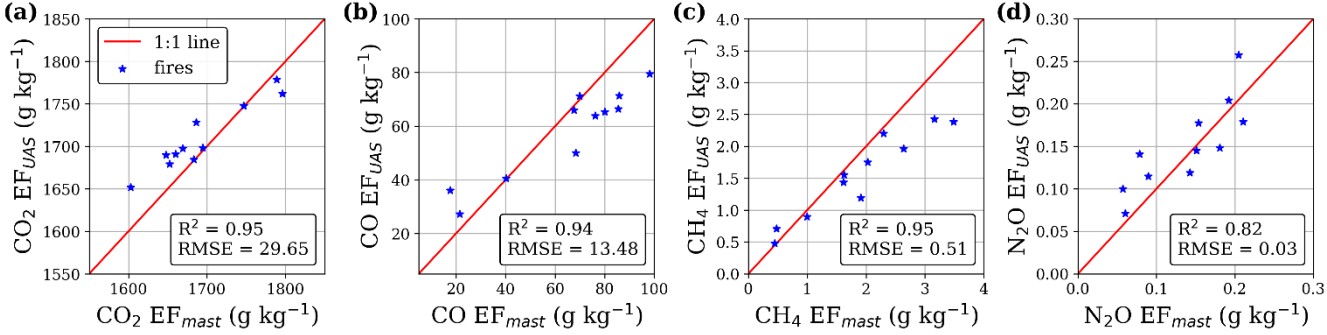

**Figure 2: Correlation between the UAS derived EF and the EF derived from the mast at the same time window for prescribed fire-experiments in the KNP. Each marker represents the EF calculated over the integrated measurements of a fire by the mast and UAS-sampled bags respectively. The 1:1 line is shown in red.**





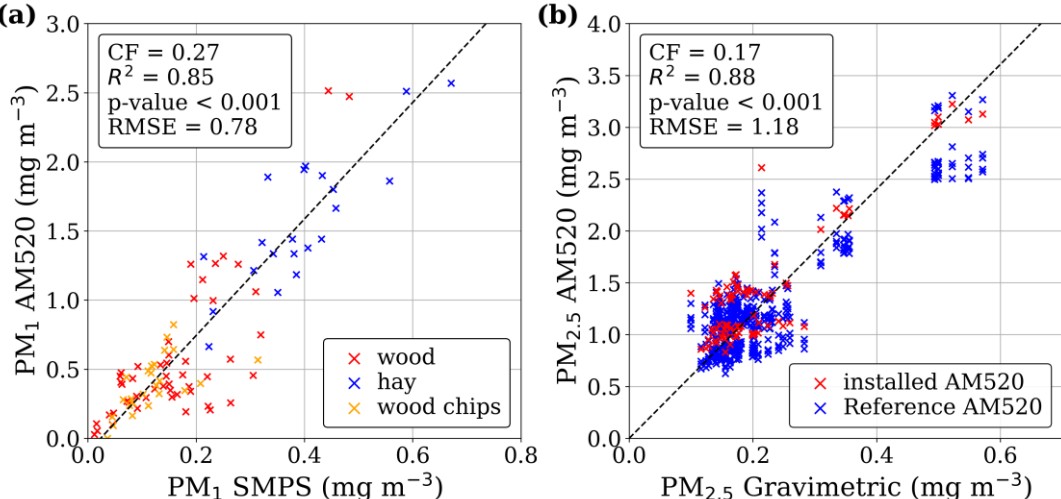

**Figure 3:** PM$_1$ concentrations from the AM520 averaged over 160 second plotted against the SMPS derived PM$_1$ measurements for the different fuel types (a). Fire-averaged PM$_{2.5}$ concentrations from 5 different AM520 modules, plotted against the gravimetric measurement from the Tapered Element Oscillating Microbalance (b). The dashed line represents the linear regression line. The VU AM520, shown in red in figure b is used for the other analysis in this study.

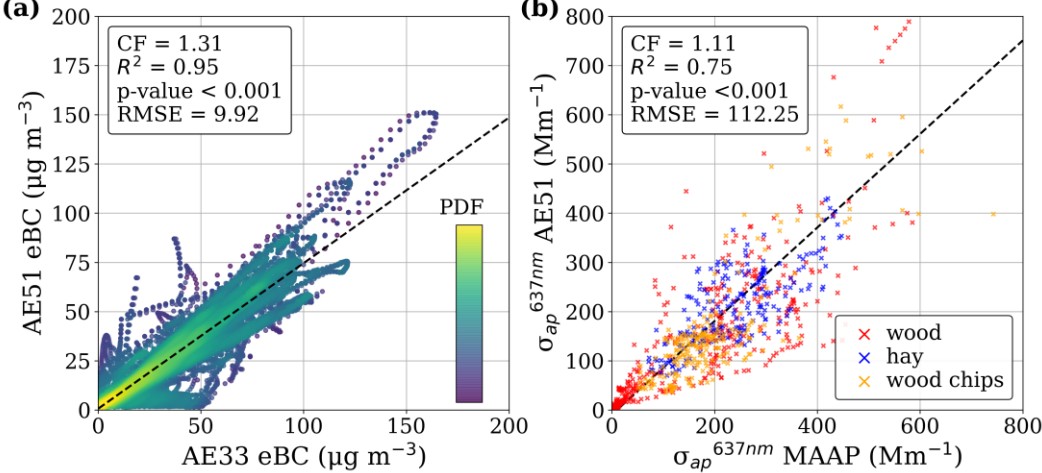

**Figure 4:** (a) Correlation between the AE51 aethalometer and the AE33 multi-wavelength aethalometer at λ = 880 nm. Colours describe the point density formula (PDF) with lighter shades representing more common values. (b) correlation between the absorption coefficient from the AE51 aethalometer (averaged over 10 seconds) and the wavelength-corrected absorption coeffiecient from the MAAP. Both figures indicate a bias, represented by the calibration factor (CF) where the AE51 slightly underestimates the absorption compared to the reference.





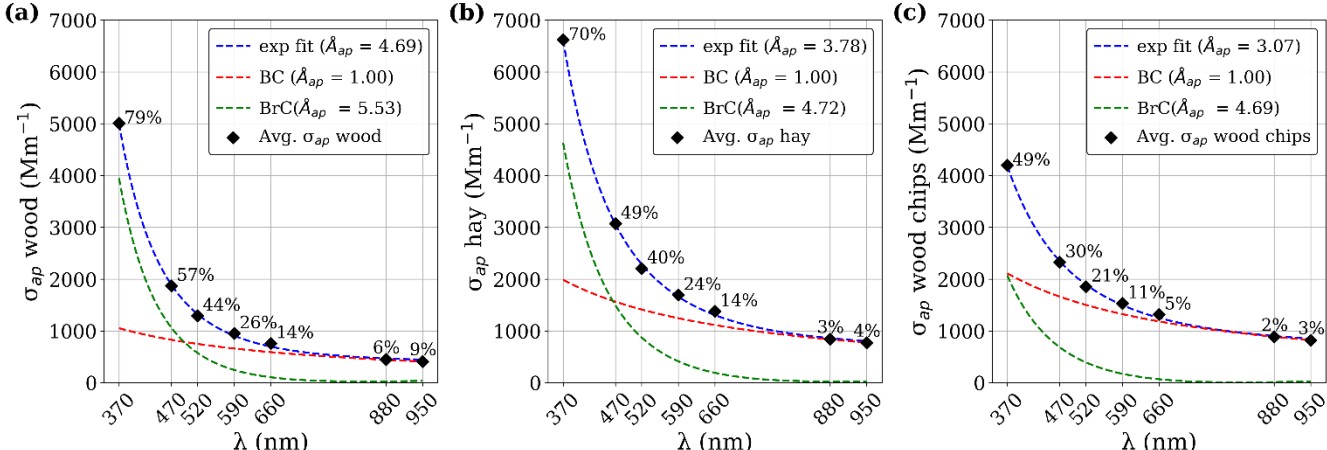

**Figure 5: Ångström coefficients based on the exponential fit through the absorption coefficients for wood (a) hay (b) and wood chips (c). The labels represent the relative contribution of BrC absorbers to the total absorption at that wavelength.**

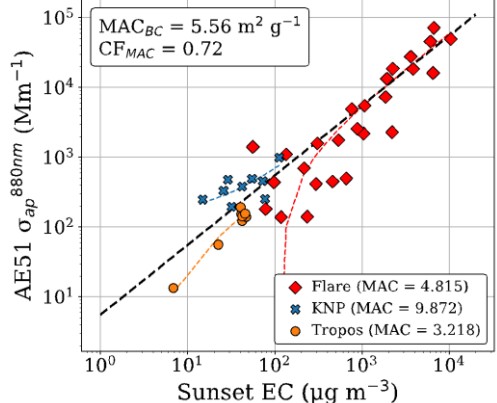

5    **Figure 6: Determination of the MAC value as the slope of the absorption coefficient at 880nm from the AE51 and the EC desorbed from the quartz-fiber filters, determined by the sunset analyser.**





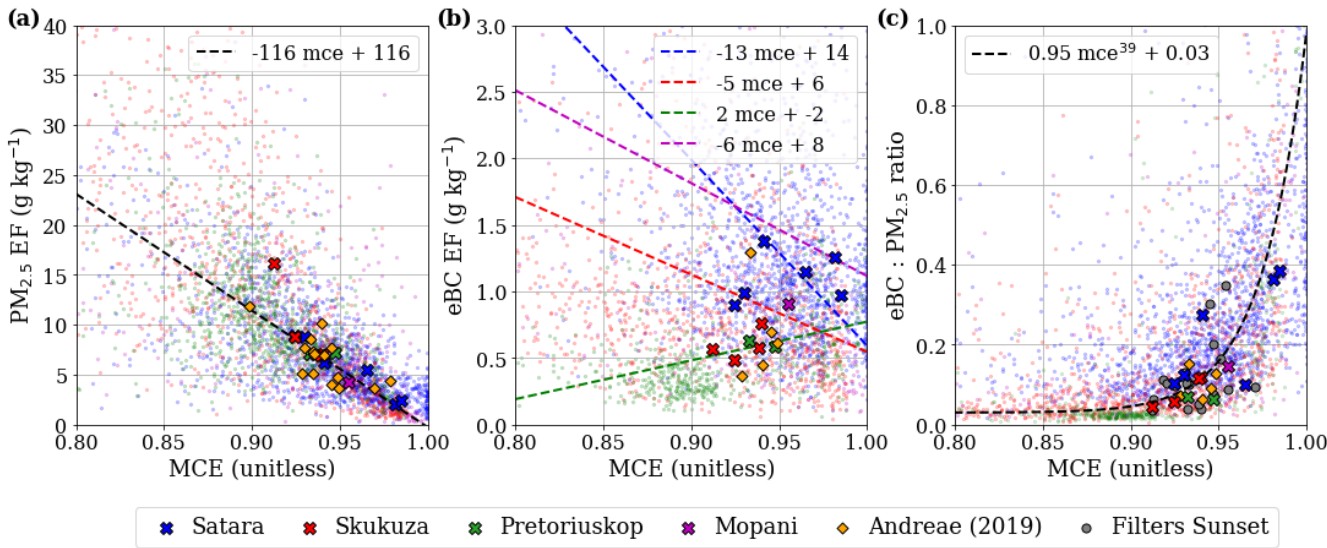

**Figure 7: Relation between the PM$_{2.5}$ EF (a) the BC EF (b) and the eBC to PM$_{2.5}$ ratio(c) with the modified combustion efficiency (MCE) in the different KNP vegetation types. Each small dot represents a 1-second measurement at the top of the mast. The large crosses represent the fire-average EF and the orange diamonds represent the study-average EF from previous savanna measurements listed by Andreae (2019). Note that none of the eBC EF regression lines are significant.**

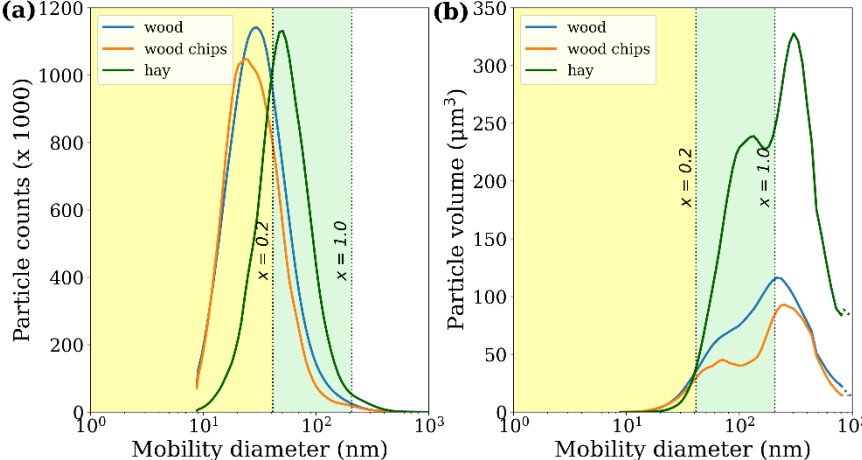

**Figure 8: Particle size distribution of the different fuel types. a) Absolute particle counts in the respective mobility diameter bins. b) The particle volume represented by the particles counted in the mobility diameter bins.**



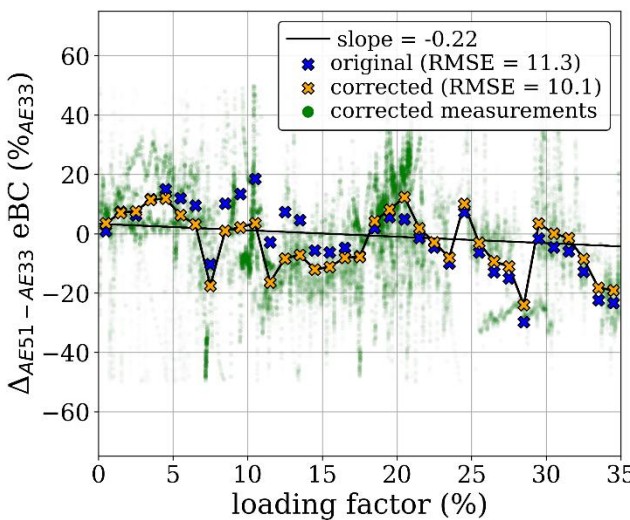

**Figure 9: Dependence of the difference between the Weingartner-corrected eBC from the AE51 and the Drinovec-corrected AE33 versus the loading factor of the AE51 filter (green dots). The blue and orange crosses represent the uncorrected and Weingartner-corrected 1% averaged differences, respectively.**

