# Peer review of "An Unmanned Aerial System (UAS) based methodology for measuring biomass burning emission factors"

_Atmospheric Measurement Techniques, 2022_

## Referee Comment (RC1)

**Anonymous Referee Comments: amt-2022-77**

**Summary**:

This manuscript presents a comparison of EFs calculated for CO, CO2, CH4, and N2O from measurements attained from a UAV (DJI Matrice 100) equipped with Tedlar bags for offline analysis with continuous measurements attained from a more heavily, traditionally-instrumented 15-m tall mast tower. In the second phase, "UAS-compatible" aerosol sensors were evaluated for accuracy and reliability of BB measurements using a series of chamber studies to estimate useful calibration factors.

**General Comments**:

The manuscript's evaluation of UAS performance and EF estimates was somewhat lacking, and instead it presented a rigorous, significant evaluation of small aerosol sensors that can potentially be deployed aboard a UAV. However, the sensors themselves were not actually outfitted for and/or deployed on an UAV platform. As it is presented, it was challenging to glean this information from the text and the title seems misleading. This would be better presented as two separate papers, with much greater detail on the UAV sampling strategy and evaluation of measurement capabilities (*e.g.*, sampling lofted versus RSC emissions, impacts of UAV prop wash, etc.) as one paper and another evaluating small aerosol sensor measurements of BB emissions. The only connection drawn between the small aerosol sensors and UAVs is that the authors mention these are "UAS-compatible," however, there is no description of power requirements, sampling time constraints, or evaluation of UAV specific turbulence/dilution impacts that could occur when deployed on a UAV. Additionally, it is not clear the UAV could carry the weight required to collect Tedlar bags as well as the aerosol sensors, therefore this is not a complete UAS BB sampling system as seemingly implied by the manuscript title. The authors rightly state that multiple carbonaceous species are necessary to calculate CMB EF, however, they don't actually present a UAV system that is capable of carrying all this instrumentation required to estimate EFs. I believe the manuscript needs to be reorganized, retitled, and instead focus only on small aerosol sensor characterization for BB sampling as significant advances to EF estimates from BB are not fully supported or described here-in.

**Specific Comments**:

The authors describe sampling biases towards RSC for ground measurements and lofted flaming emissions for airborne sampling. The UAV sampling approach described measures at the height of the stationary mast to compare with. While these comparisons are useful to evaluate the onboard drone measurements, it would be interesting to show the utility of UAVs by sampling different areas of the plume and evaluating varying emissions for their dependences on burn/ environmental conditions. How do these differences compare to laboratory and ground/air measurements? What new information is available by sampling with the UAV? We obviously aren't attaining more speciated emissions, but are these EFs useful and what can they additionally provide compared to more traditional fire-integrated biome specific EFs used in inventories?

P3 L16 You mention "Better understanding of this BB EF variability would improve our quantification of fire emissions, and would aid understanding of the effects of future climate- and human-induced changes in fire regimes." -- How does knowing the BB EF variability actually improve upon our current fire emissions estimates? There are currently no methods to implement time-varying, burn condition (MCE) varying EFs. Additionally, the biome-average EF is shown to be one of the smaller uncertainties when estimating total emissions from fires (*i.e.*, fuel consumption and burned area estimates have larger uncertainties). I would like to see stronger support describing the importance/motivation for UAV-based EF measurements, especially since the methods described don't provide any detailed speciation or spatial plume variability.

P5 L26 Did you use an onboard pump to fill the bag? The UAV experimental details are lacking in this manuscript. It is insufficient to refer readers to sampling details outlined in Vernooij et al., 2021 since the title of this manuscript is UAS-methodology. What are the flight times and flight restrictions (wind, instrument load, power draw)?

How does airflow/downwash from propellers impact dilution of smoke sample? How would this turbulence and dilution impact your aerosol optical properties or loading? Would gas-phase and particle-phase collection be impacted by dilution differently? What strategies are necessary when setting up sampling inlets for both on a UAV with different flow rates?

P6 The authors assume a total amount of carbon emitted using literature reported NMHC and carbon-containing particles, shouldn't this vary depending on whether you were sampling a high or low MCE fire? What uncertainties are associated with assuming this is constant across the entire sampled fire or for different fuels and on different days? Are you biased or limiting the measurement variability you observe by assuming a constant carbon contribution from these species?

P2 L7 I wouldn't say BB is a "main source" of GHGs, especially with the CO2 sequestration you mention in the following sentences. A slight rewording would be nice.

P3 L4 The authors only mention burned area based approaches, you could also mention FRP-based approaches since many still rely on EFs

P3 L20 Many more recent references could be added. As an example, Yokelson has used a land-based cart for ground measurements for many years. Also, the way this is phrased implies there have not been many studies investigating EFs, and even recently there are many BB-focused airborne campaigns in the U.S. (WE-CAN, FIREX-AQ, BBOP, SEAC4RS) and internationally that could be mentioned.

P3 L23-26, The authors question the applicability of aerosol EFs from laboratory measurements, due to the uncertain evolution of aerosols, however, an emission factor has nothing to do with evolution and is by definition the amount emitted by the actual fire, not the amount following transformation and ageing.

P3 L32 What do you mean by "fire products?" Do you mean the distribution of emissions is not equally distributed over all areas of a smoke plume. Please clarify this sentence

Figure 1, can you show the stationary mast average mixing ratio over the UAS sampling time? You show the EFs correlate across the entire fire sampling interval in Figure 2, so I assume they mostly agree though it'd be good to see the absolute difference. You can also shorten the x-axis since there is no detail or additional UAS sampling beyond 15 minutes since ignition.

P11 L10 what do you mean by cumulative emissions? Does this mean you sum the emissions across all sampled bags in a single fire to get one value per fire?

To derive CFs you use hay, wood, and wood chips, why not just burn savanna grasses. Hay seems most similar, but I'd expect some differences. Do you even use the chamber experiments that burned peat and straw anywhere else in this manuscript?

It isn't clear why you have three separate chamber experiments. Maybe detail what the usefulness and differences are between each chamber experiment and what the science foci of each set of experiments were.

P12 L25 should you reference Fig 3b here?

Fig 4 Is this for multiple fires?

Fig 6 What is your justification for using the average of all to estimate a CF rather than just using the field (KNP) average value?

Fig 7 What are the fits shown? For instance, in plot (b) the red dashed line doesn't seem to be the fit for the red crosses.

P14 L26-32 This paragraph seems out of place and doesn't tie to your measurements well.

I think some reorganization is necessary. For instance, sect 4.2 seems out of place and instrument performance should move to the methods section. I needed to jump around frequently to try and follow UAV measurements and chamber measurements. The mix of aerosol sensor/chamber analysis versus UAV sampling is mixed in an unorganized fashion.

P18L30 "for variability within individual fires was difficult as separate filters for smaller periods of the fire (e.g. the flaming and smouldering phase) resulted in insufficient filter loading." This is important to highlight in more areas of the text as this would have important implications when sampling with a UAV-based system.

**Technical corrections**

P3 L9 "usuall calculated as from" should be "usually calculated from"

P3 L11 "is dependent on weather conditions and fuel characteristics" should also list "burn conditions"
P3 L15 EF variability "show high intra-biome variability" This should be less than 30%, which is small relative to larger burned area uncertainties

Table 2 is mentioned in text before Table 1

P4 L3 "maybe" should be "may be"

P4 L4 (airplane),?

P4 L5 "maybe" should be "may be"; also misspelled "measurements"

P12 L13 "and the gravimetrically" -rephrase

P13 L7 "wether" should be "whether"

P20 L30 change "where" to "here"

---

## Referee Comment (RC2)

This manuscript developed a lightweight UAS sampling system for the measurements of pollution emission factors from biomass burning. Gas emissions were collected by Tedlar bag for offline analysis, while PM and BC were measured by sensors onboard UAS. Sensor performance was assessed by comparing the sensor results to those from high-fidelity equipment in the laboratory smoke experiments. And the performance of the UAS system was evaluated by comparing to co-located mast measurements for prescribed burning experiments in the Kruger national park in South Africa. Combining laboratory characterization and field measurements, the authors demonstrated that with proper correction factors applied, the UAS system can serve as a promising tool for obtaining representative biomass burning emission factors. Overall, the manuscript is well written. The techniques proposed are valuable to the literature. I recommend the manuscript be considered for publication only after my following comments are fully addressed.

**General comments:**

1. Title: For the UAS system, there are quite a lot of differences between fixed-wing and copter-type UASs with regard to factors such as payload, battery limitation, propeller influence, and sensor integration. Until reading the Method section, I realized that this study used a copter-type UAS (DJI Matrice 100). I suggest that the authors mention this information directly in the title. The title can be modified as, for example, "A copter-type unmanned aerial system based methodology for measuring biomass burning emission factors"

2. Page 3 line 34: "Most atmospheric models account for photochemical processing, but not the chemical changes associated with the initial cooling of the smoke to ambient temperature."

   What types of chemical changes are associated with the initial cooling of the smoke? Please clarify. How can UAS-enable technique help to address this issue?

3. Section 2.1.2: A picture of the Mast and UAS setup would be helpful for the readers to gain a better idea of the measurement design (e.g., mast height, gas inlet of the mast, locations of equipment and sensors on mast, and locations of Tedlar bag and sensors on UAS).

4. Page 5 lines 26-31: This paragraph is apparently too short, as this manuscript focuses on UAS-based measurement. The description regarding UAS sampling needs to be significantly improved. Although the authors mentioned that detailed methodology can be found in Vernooij et al. (**2021**), there are only two short paragraphs presented in that paper.

   Questions to clarify:

   a.  Why DJI M100 was selected? What was the payload limit of DJI M100?

b. How was the Tadlar bag mounted to the UAS? How was gas sampling performed? Was a pump used during the sampling? Was it a Teflon pump? What was the flow rate of the sampling? How much air was collected in each bag? Was the flow rate recorded during the sampling? Was temperature recorded during the sampling for concentration correction? Was sampling quality control well performed (e.g., no leaking)? Samples were analyzed within 12 hours of sampling. How were the samples stored? Did it affect the concentration of the species inside the bag?

c. What were the weights of the AE51 and AM520? Given the payload, how long was each flight? Was the sampling inlet position on the top of UAS or below UAS? Were the data of AE51 and AM520 transmitted to the UAS controller at real time? Or they were stored in SD cards and retrieved after each flight?

d. Were UAS gas sampling and PM measurement influenced by the propellers? What were the influence lengths of the UAS-induced wind field above and below the UAS? Did it affect the mixing of biomass burning emissions and thus impact the estimation of emission factors?

5. Page 5 line 27: UAS sampling was conducted at an altitude of 15 m, similar as the sampling altitude at the mast. Why was 15 m selected for emission factor calculation? Please clarify.

6. Page 6 line 12: "For the bag and mast measurements, we used the PM to CO ratio based on AM520 and CRDS measurements, with carbon accounting for 68% of the PM-mass (Reid et al., **2005a**)."

   This sentence is confusing. I understand that PM to CO ratio was calculated based on AM520 and CRDS. How was the PM carbon fraction calculated? Did you calculate the carbon fraction? Or it was obtained from Reid et al. (**2005**). If the fraction was obtained from the literature, why this value was selected? How representative is this value?

7. Page 7 lines 20-23 and Page 13 lines 3-5: as the authors mentioned, one of the reasons that AE51 is not as accurate as AE33, especially at low concentrations, is that AE51 operates at a much lower flow rate. One suggestion for future improvement can be increasing the sampling flow rate of AE51 using an external pump, as demonstrated by Wu et al. (*Science of the Total Environment*, **2021**).

   Reference: Wu et al., Vertical profiling of black carbon and ozone using a multicopter unmanned aerial vehicle (UAV) in urban Shenzhen of South China, *Science of the Total Environment*, **2021**, *801*, 149689.

8. Page 11 lines 5-6: "After the flaming phase ceased, mixing ratios and thus temporal varying EFs (green lines) for CO, $CH_4$ and $PM_{2.5}$ sharply rose."

What do you mean by "After the flaming phase ceased"? Do you mean after around 3 min in Figure 1? The emission factors of CO, $CH_4$, and $PM_{2.5}$ increased sharply after 3 min in Figure 1. However, why no increases in emissions were observed? Instead, the emissions decreased significantly after 3 min.

9. Page 16 line 32: "concentration. Cross-correlation of the UAV-mounted AM520 to five co-located AM520 modules revealed measurement errors of up to 20%."

   Can the authors provide the figures showing the good correlation between the datasets collected from AM520 onboard UAV and those from AM520 on the mast? This can be good supporting information demonstrating the validity of UAS data.

10. The authors did a very good job discussing the assumptions and the uncertainties of this study. It would also be valuable to the readers if the authors can provide some recommendations (or improvements) for future studies using the UAS system for emission factor measurement.

**Technical comments:**

1. Page 2 line 25: change "different fuel types" to "different fuel type"

2. Page 3 line 9: typo "usuall"

3. Page 4 line 5: typo "measuremetns"

4. Table 2 is shown and discussed before Table 1 in the manuscript. Please adjust the order.

5. Page 9 equation 8: "MAAP" not "MAAB"

6. Page 10 line 18: delete "with a"

7. Page 13 line 7: typo "wether"

8. Page 13 line 20: Fig. 8? There is no Fig. 8. Do you mean Fig. 6?

9. Page 20 line 30: "Such FRP measurements were not available where, and although we measured the atmospheric concentrations continuously as the different stages…"

   Missing information after "where"

---

## Author Response (AR1)

**Response of the authors to comments by reviewers – "An Unmanned Aerial System (UAS) based methodology for measuring biomass burning emission factors"**

Roland Vernooij (corresponding author) on behalf of the authors:

We thank both reviewers and the editor for their time and effort in assessing our manuscript, and the detailed and constructive comments which helped to improve the quality and clarity of this paper. Please find below our point-to-point response to the review. The revised text and updated figures are included in the updated manuscript. A separate 'track-changes' document is included to highlight the changes that were made to the manuscript. The text in italics below refers to the original reviewer remarks.

**Reviewer # 1 general comments**

*The manuscript's evaluation of UAS performance and EF estimates was somewhat lacking, and instead it presented a rigorous, significant evaluation of small aerosol sensors that can potentially be deployed aboard a UAV. However, the sensors themselves were not actually outfitted for and/or deployed on an UAV platform. As it is presented, it was challenging to glean this information from the text and the title seems misleading. This would be better presented as two separate papers, with much greater detail on the UAV sampling strategy and evaluation of measurement capabilities (e.g., sampling lofted versus RSC emissions, impacts of UAV prop wash, etc.) as one paper and another evaluating small aerosol sensor measurements of BB emissions. The only connection drawn between the small aerosol sensors and UAVs is that the authors mention these are "UAS-compatible," however, there is no description of power requirements, sampling time constraints, or evaluation of UAV specific turbulence/dilution impacts that could occur when deployed on a UAV. Additionally, it is not clear the UAV could carry the weight required to collect Tedlar bags as well as the aerosol sensors, therefore this is not a complete UAS BB sampling system as seemingly implied by the manuscript title. The authors rightly state that multiple carbonaceous species are necessary to calculate CMB EF, however, they don't actually present a UAV system that is capable of carrying all this instrumentation required to estimate EFs. I believe the manuscript needs to be reorganized, retitled, and instead focus only on small aerosol sensor characterization for BB sampling as significant advances to EF estimates from BB are not fully supported or described here-in.*

**Authors response to Reviewer # 1 general comments**

We thank the reviewer for the thorough and constructive feedback. We recognize the criticism and realize the description of the UAS system was insufficient and easily led to misunderstanding. We have added a new Figure (see below), added more detailed information on the UAS-system, and clarified several sections to better show how the different parts are linked. We hope this remedies most of the issues brought up by the reviewer. Below is the new Figure (Fig. 1).

[Figure]

*Figure 1: Photo of the UAS and schematic representation of the gas sampling system (blue) and aerosol measurement system (black) as installed on the DJI Matrice 200 UAS.*

We clarified several sections as shown in the document with track changes, for example in the methods section we added: 'The combined set-up equipped on a Matrice 200, takes collocated measurements of the $CO_2$, CO, $CH_4$

and $N_2O$ mixing ratio, as well as the $PM_{2.5}$ and eBC mass concentration, and compute the BB EFs of these species. While an earlier version of the UAS in Fig.1 (the DJI Matrice 100) was used for some of the experiments described in this study, it has insufficient payload capability for the combined system.' We describe how the proposed set-up deals with the reviewer's concerns (e.g. sampling strategy in much greater detail), which we feel has significantly improved the manuscript. We have also added additional motivation on how UAS-based EF measurements may contribute to our knowledge of EF variability and eventually benefit emissions estimates. In our opinion, while progress will be made through a more detailed coverage of species, which indeed may prove easier from larger aircraft or ground measurements, but also through a reduced sampling bias towards smouldering or flaming emissions and a better spatiotemporal coverage of a fire. We feel this is one of the most significant contributions of our paper; it shows the feasibility of this approach.

We have considered to separate the text into two papers, following the reviewer's suggestion. However, since the aerosol and gas sensors are in fact both co-located on the proposed UAV, it describes a complete UAS BB sampling system that allows for EF measurements. The gas and aerosol measurements have different strengths and weaknesses. While the gas analyzers are well accepted, the applicability of the bag-sampling approach still needed to be demonstrated. In the aerosols case, we have direct online measurements but the performance for BB aerosols needs to be proven. While this requires different experiments and a slightly more complex paper, we feel that the strength is in the combination of the measurements into a single system. Splitting the manuscript into two papers would in our view result in one overly short paper, a lot of duplicate introductory text and an increased burden on reviewers.

| Reviewer # 1 detailed comments | Author's response, reasoning and comments |
|---|---|
| P3 L16 You mention "Better understanding of this BB EF variability would improve our quantification of fire emissions, and would aid understanding of the effects of future climate- and human-induced changes in fire regimes." -- How does knowing the BB EF variability actually improve upon our current fire emissions estimates? There are currently no methods to implement time-varying, burn condition (MCE) varying EFs. Additionally, the biome-average EF is shown to be one of the smaller uncertainties when estimating total emissions from fires (*i.e.*, fuel consumption and burned area estimates have larger uncertainties). | The acquisition of a large dataset of in-situ EF measurements with a large spatio-temporal coverage is a first step and prerequisite for the introduction of variable EFs in global models. The datasets we are collecting will be the basis for a spatio-temporal assessment of emission factors in savannas used in GFED5 (Vernooij et al., in preparation). We fully agree that uncertainties in other factors of emission estimation may be larger, but are also confident we can significantly improve on the current approach with only one static emission factor per biome. For example, models currently strongly overestimate the $CH_4$ and CO emissions from xeric savannas while underestimating emissions from woodland savannas.

In P3L16 we changed 'high intra-biome variability' to 'substantial intra-biome variability' |
| I would like to see stronger support describing the importance/motivation for UAV-based EF measurements, especially since the methods described don't provide any detailed speciation or spatial plume variability. | UAVs tackle some of the limitations of other methods in terms of biases towards small prescribed fires, or smouldering/flaming emissions. Also, their versatility makes them very suitable to follow more and larger fires for longer periods.

To illustrate this versatility: Since the development of the proposed methodology, we have collected over 4000 gas samples covering many different savanna ecosystems in South America, Africa and Australia (e.g. Vernooij et al., 2021; Russel-smith et al. 2021).

We added the following sentence (p4 L10):

'UAS-systems offer a low-cost and versatile solution for sampling a mixture of flaming and RSC emissions within a freshly emitted, dense smoke |

| | plume (Aurell et al., 2021; Vernooij et al., 2021). The flexibility of a UAS tackles some of the major weaknesses of ground and mast measurements. The system can be quickly deployed when a fire is sighted, eliminating the bias towards small "experimental plot" fires. Also, by measuring a fire for hours burning through large swaths of vegetation, the UAS provides much better spatio-temporal coverage of the fire.' |
|---|---|
| P5 L26 Did you use an onboard pump to fill the bag? The UAV experimental details are lacking in this manuscript. It is insufficient to refer readers to sampling details outlined in Vernooij et al., 2021 since the title of this manuscript is UAS-methodology. What are the flight times and flight restrictions (wind, instrument load, power draw)? | Thank you for pointing out the lack of detail in the set-up. We have added a figure and description to illustrate the proposed measurement set-up at the start of the methods section:

"Figure 1 shows a schematic overview of the proposed measurement systems installed on the UAS (DJI Matrice 200). To prevent rotor-induced pressure alterations affecting aerosol mass concentrations, the inlets were extended using a carbonfibre tube to outside the rotor affected area. The gas-sampling system (530g) consists of an Arduino operated remote control and logging module, a 1.6 L min$^{-1}$ diaphragm gas pump (NMP 015, KNF), a four-way manifold connected to four 1L Tedlar bags and a carbonfibre cage (Fig 1., schematics in blue). This setup was based on an earlier setup deployed on a DJI Matrice 100 UAS described in Vernooij et al. (2020), but included a flushing mechanism to flush the extended inlet before sampling. The aerosol sampling system (Fig 1, schematics in black) contains two inlets. The first inlet is fitted with an inertial impactor (Personal Modular Impactor, SKC) followed by a 37mm quartz-fibre filter (Tissuquartz 2500QAT-UP, Merck) and a flow-controlled pump. The filtered air is used to dilute the stream coming from the second inlet using a 1:1 ratio to prevent saturation of the AM520 and limit the filter loading effect in the AE51. All tubing is polyurethane and kept as short as possible, whilst avoiding sharp corners. PM2.5 and eBC are continually logged in the AM520 and AE51 respectively. For each sample, the start time, end time, temperature, pressure and relative humidity at the UAS are logged. The transport time from the inlet to the measurement equipment is corrected for when computing EFs. The overall flight time of the system is roughly 15 minutes for each set of TB50 batteries, which is enough to fill 12-16 bags. The combined set-up equipped on a Matrice 200, takes collocated measurements of the $CO_2$, CO, $CH_4$ and $N_2O$ mixing ratio, as well as the $PM_{2.5}$ and eBC mass concentration, and compute the BB EFs of these species. While an earlier version of the UAS in Fig.1 (the DJI Matrice 100) was used for some of the experiments described in this study, this has insufficient payload capability for the combined system." |

| | |
|---|---|
| How does airflow/downwash from propellers impact dilution of smoke sample? How would this turbulence and dilution impact your aerosol optical properties or loading? Would gas-phase and particle-phase collection be impacted by dilution differently? What strategies are necessary when setting up sampling inlets for both on a UAV with different flow rates? | Downflow from the propellers may indeed impair the particle measurements by affecting the air pressure and velocity at which particles pass the pump inlet. To remedy this, we therefore extended the inlets to outside of the rotor airflow.

In the proposed setup, there are two separate collocated inlets for gasses and aerosols. The dilution only affects the aerosol stream as is depicted in Figure 1. The delay in the measurement (from inlet to measurement) is then accounted for in the script when calculating the EFs. |
| P6 The authors assume a total amount of carbon emitted using literature reported NMHC and carbon-containing particles, shouldn't this vary depending on whether you were sampling a high or low MCE fire? What uncertainties are associated with assuming this is constant across the entire sampled fire or for different fuels and on different days? Are you biased or limiting the measurement variability you observe by assuming a constant carbon contribution from these species? | We assumed the ratio of carbon in NMHC to carbon in $CH_4$ was constant, based on previous savanna measurements that measured both. When PM was not measured, we also assumed the ratio of carbon in particulates to be proportional to the CO emissions based on the ratio derived from previous studies.

This does not mean the absolute amount is constant but rather the ratio. Using these assumed relationships, the total amount in NMHC and carbon-containing particles is therefore indeed dependent on the MCE of the fire. Overall, the carbon in PM and NMHC constitute respectively 0.5−2% and 0.4−3% of the total emitted carbon. Therefore, the uncertainty from the effect this assumption on the EFs of gaseous species is limited.

We changed the text to:

'Because we did not measure the non-methane hydrocarbons and the chemical composition of carbonaceous particulates, the NMHC and the carbon content of the particulates was estimated based on literature values in order to estimate $C_{total}$; The total amount of carbon in non-methane hydrocarbons was estimated to be 3.5 times the $ER(CH_4/CO_2)$ based on common ratios for savanna fires (Andreae, 2019; Yokelson et al., 2011, 2013). For the bag and mast measurements, we used the PM to CO ratio based on AM520 and CRDS measurements, with carbon accounting for 68% of the PM-mass (Reid et al., 2005a). Overall, the carbon in PM and NMHC constitute respectively 0.5−2% and 0.4−3% of the total emitted carbon. Therefore, the uncertainty from this assumption on the EFs of gaseous species is limited.' |
| P2 L7 I wouldn't say BB is a "main source" of GHGs, especially with the $CO_2$ sequestration you mention in the following sentences. A slight rewording would be nice. | We changed the sentence to: 'Landscape fires, also referred to as biomass burning (BB), are a substantial source of GHG and aerosol emissions to the atmosphere. |
| P3 L4 The authors only mention burned area based approaches, you could also mention FRP-based approaches since many still rely on EFs | We agree and changed the sentence to: 'BB emission inventories are used to study the impact of fires on regional and global biogeochemical cycles. In these inventories, emissions are generally |

| | calculated based on the consumed fuel (either calculated though a modelled fuel load and satellite derived burned area, or though satellite measurements of fire radiative power integrated over time) and field measurements of emission factors (EF) (Seiler and Crutzen, 1980).' |
|---|---|
| P3 L20 Many more recent references could be added. As an example, Yokelson has used a land-based cart for ground measurements for many years. Also, the way this is phrased implies there have not been many studies investigating EFs, and even recently there are many BB-focused airborne campaigns in the U.S. (WE-CAN, FIREX-AQ, BBOP, SEAC4RS) and internationally that could be mentioned. | We appreciate the comment and agree our wording lacks appreciation for all those efforts. We have revised the text and added more recent references. 'Field measurements of BB EF have been derived for a wide variety of vegetation types and species using data from in situ sensors carried on the ground (e.g. Zhang et al., 2015; Wooster et al., 2018; Reisen et al., 2018), mounted on masts (e.g. Korontzi et al., 2003; Wiggins et al., 2020), or aircraft (e.g. Liu et al., 2017; May et al., 2014; Yokelson et al., 2007; Barker et al., 2020; Thompson et al., 2020)' |
| P3 L23-26, The authors question the applicability of aerosol EFs from laboratory measurements, due to the uncertain evolution of aerosols, however, an emission factor has nothing to do with evolution and is by definition the amount emitted by the actual fire, not the amount following transformation and ageing. | We agree and have changed our wording again. What we meant is that previous comparisons of laboratory versus field derived EFs from the same fuel (e.g. Yokelson et al. 2013) show significant differences between the two and therefore care should be taken when using laboratory studies to resemble field conditions. We revised the text in P3L28 to: 'Many laboratory studies have examined EFs during indoor experiments including those looking at the characterization of BB particulate emissions (Reid et al., 2005a, Yokelson et al., 2013). However, the representativeness of these measurements to natural fires is uncertain, considering that important field conditions affecting EFs, e.g. wind, fuel moisture content, fuel structure and temperature, are difficult to include in the experiments. This generally leads to higher combustion efficiency in a laboratory setting (Liu et al., 2014, 2017; May et al., 2014; Yokelson et al., 2013; Thompson et al., 2020). Airplane measurements also require additional information before they resemble the actual field EF's given that the optical and chemical properties of BB aerosols change with the ageing of the smoke (Cappa et al., 2020; Pokhrel et al., 2016; Vakkari et al., 2014). Differences in atmospheric lifetime, hygroscopic growth, coating of soot by OC, and susceptibility to vertical and lateral transport, all complicate EF comparisons made at different points downwind (Adachi et al., 2010).' |
| P3 L32 What do you mean by "fire products?" Do you mean the distribution of emissions is not equally distributed over all areas of a smoke plume. Please clarify this sentence | By fire products we meant emissions from the RSC and flaming phases. To clarify this, we changed the text in P4L8 to: 'EFs should represent a mixture of emissions from the smouldering and flaming phases (Akagi et al., 2013; Ward and Radke, 1993). Aircraft measurements may be biased towards flaming emissions, since they sample lofted emissions that |

| | typically result from higher intensity combustion, whereas ground measurements may be biased towards residual smouldering combustion (RSC) emitted species since the smoke from higher intensity burns is lofted out of reach.' |
|---|---|
| Figure 1, can you show the stationary mast average mixing ratio over the UAS sampling time? You show the EFs correlate across the entire fire sampling interval in Figure 2, so I assume they mostly agree though it'd be good to see the absolute difference. You can also shorten the x-axis since there is no detail or additional UAS sampling beyond 15 minutes since ignition. | We agree that the X-axis could be shortened as the emissions become insignificant after 15 minutes. However, this notion is important; a major concern we had was that if RSC emissions would remain for an extended period of time, the UAS methodology would not be more versatile compared to a static mast.

 We have now added the mast average mixing ratio in Figure 1 in blue. On average in this profile, the absolute difference between the bags and the mast were less than 5%, which is low concerning the location is not exactly the same. |
| P11 L10 what do you mean by cumulative emissions? Does this mean you sum the emissions across all sampled bags in a single fire to get one value per fire? | This is correct. By calculating the EFs over the cumulative emissions rather than taking the average EFs over the bags, the bags are weighed by the excess mixing ratio in each bag to get the fire weighted average (WA) EF.

 We clarified this in the text P12L25: 'Fig. 3 represents the WA EF from the UAS-sampled bags, calculated based on the sum of the emissions across all sampled bags in a single fire, plotted against EFs calculated from the cumulative emissions that passed the mast with each point representing a single fire (11 fires in total).' |
| To derive CFs you use hay, wood, and wood chips, why not just burn savanna grasses. Hay seems most similar, but I'd expect some differences. Do you even use the chamber experiments that burned peat and straw anywhere else in this manuscript? | Peat and straw fuels were used for the calibration of the AM520 against gravimetrical equipment and cross calibration (Fig. 4b). We have added this to the caption:

 'Fire-averaged $PM_{2.5}$ concentrations from 5 different AM520 modules during 10 experiments burning peat and straw, plotted against the gravimetric measurement from the Tapered Element Oscillating Microbalance'

 In hindsight we should have used grasses and thus agree with the reviewer and keep this for future work. The advantage of using wood and wood chips is that our results may be useful when studying other biomes than savannas. |
| It isn't clear why you have three separate chamber experiments. Maybe detail what the usefulness and differences are between each chamber experiment and what the science foci of each set of experiments were. | The manuscript describes a progression of follow-up experiments, each with its own merits and flaws. The next step will be to build on this and start several field experiments where all instruments will be used as a single system. We have aimed to better describe this progression to avoid confusion. For example, we revised the text in P7L13 to:

 **'2.3 Aerosol calibration experiments**

 For the second phase, we performed BB experiments in the Leipzig aerosol chamber at the |

| | Leibniz Institute for tropospheric research (TROPOS), the Kings Wildfire Testing Chamber (KWTC) in London, and the Fire Laboratory of Amsterdam for Research in Ecology (FLARE) where we calibrated the mobile aerosol analysers against different types of high-fidelity laboratory equipment. At TROPOS and FLARE, wooden logs, wood chips, and hay were burned in an actively vented combustion chamber, connected to the measurement equipment. The experiments conducted at FLARE served to compare the AE51 and AM520 BC/PM ratios to the EC/OC ratios determined by the Sunset analyser and perform a recalibration of the MAC-value. The experiments at TROPOS served to compare the AE51 to the AE33 and the MAAP during biomass burning experiments as well as to compare the AM520 to SMPS particle counts. At the KWTC, smoke from peat fires and straw was allowed to stabilize around predetermined levels in a smoke chamber which was connected to a series of analysers. The experiments at KWTC served as a direct comparison of the mass concentration obtained by the AM520 (which uses assumptions for density and particle size) with gravimetrically obtained mass concentrations. Additionally, we performed an inter-comparison between 6 AM520 modules.' |
|---|---|
| P12 L25 should you reference Fig 3b here? | Thank you, we added the reference to the figure |
| Fig 4 Is this for multiple fires? | Correct, Figure 4A shows 14 fires at the Leibniz Institute for tropospheric research (TROPOS), Leipzig. For fig 4B, 10 experiments at the Kings Wildfire Testing Chamber (KWTC), London are shown. This information is now added to the caption. |
| Fig 6 What is your justification for using the average of all to estimate a CF rather than just using the field (KNP) average value? | We agree that using a specific MAC value for the field measurements is ideal. We have recalculated the BC EFs based on only the KNP values and revised Figure 8. |
| Fig 7 What are the fits shown? For instance, in plot (b) the red dashed line doesn't seem to be the fit for the red crosses. | The dashed lines indeed show the fits for the various vegetation types. However, since none of them is significant and may lead to confusions we have removed this information from the figure. |
| P14 L26-32 This paragraph seems out of place and doesn't tie to your measurements well. | We agree that the paragraph is a bit off topic and unnecessary, and have therefore removed it from the discussion. |
| I think some reorganization is necessary. For instance, sect 4.2 seems out of place and instrument performance should move to the methods section. I needed to jump around frequently to try and follow UAV measurements and chamber measurements. The mix of aerosol sensor/chamber analysis versus UAV sampling is mixed in an unorganized fashion. | Based on the reviewer's advice we have tried to restructure the manuscript. While we agree that the manuscript can require some jumping around, our attempts at restructuring did not improve readability in our view. We restructured the manuscript several times before we reached the present format. Since the discussion of the performance in sect. 4.2 is based on our own results (described in sect. 3.2 and 3.3) and literature comparisons, we feel it does belong with the discussion rather than the methods. |

| | By better explaining the structure upfront we hope we now have a easier to read paper. |
|---|---|
| P18L30 "for variability within individual fires was difficult as separate filters for smaller periods of the fire (e.g. the flaming and smouldering phase) resulted in insufficient filter loading." This is important to highlight in more areas of the text as this would have important implications when sampling with a UAV-based system. | To clarify the importance of this, the filters serve as an individual fire recalibration of the MAC-value and provide important insights on their own. However, if there is insufficient loading on the filter to perform this recalibration, the average MAC-value of similar fires may be used. Although we identify this recalibration as being important, it is not done by other studies.

Also, rather than being specific to this UAS-method, this would be a lot more problematic in for instance aircraft-based methods which typically operate in much more dilute smoke. |
| Technical corrections: See document | Thank you, the technical corrections were corrected accordingly in the revised document and are highlighted in the included 'track changes' document. |

**Reviewer # 2**

*This manuscript developed a lightweight UAS sampling system for the measurements of pollution emission factors from biomass burning. Gas emissions were collected by Tedlar bag for offline analysis, while PM and BC were measured by sensors onboard UAS. Sensor performance was assessed by comparing the sensor results to those from high-fidelity equipment in the laboratory smoke experiments. And the performance of the UAS system was evaluated by comparing to co-located mast measurements for prescribed burning experiments in the Kruger national park in South Africa. Combining laboratory characterization and field measurements, the authors demonstrated that with proper correction factors applied, the UAS system can serve as a promising tool for obtaining representative biomass burning emission factors. Overall, the manuscript is well written. The techniques proposed are valuable to the literature. I recommend the manuscript be considered for publication only after my following comments are fully addressed.*

**Authors response to Reviewer # 2 general comments**

Many thanks for the review and constructive criticism and thorough review. We hope that with the proposed revision, we sufficiently address your concerns and the manuscript could be considered for publication. Below we address the detailed comments individually. The technical corrections are addressed accordingly and are marked in the 'track-changes' document.

| Reviewer # 2 detailed comments | Author's response, reasoning and comments |
|---|---|
| Title: For the UAS system, there are quite a lot of differences between fixed-wing and copter-type UASs with regard to factors such as payload, battery limitation, propeller influence, and sensor integration. Until reading the Method section, I realized that this study used a copter-type UAS (DJI Matrice 100). I suggest that the authors mention this information directly in the title. The title can be modified as, for example, "A copter-type unmanned aerial system-based methodology for measuring biomass burning emission factors" | Thank you for the suggestion, to clarify this we have changed the title to 'A quadcopter Unmanned Aerial System (UAS) based methodology for measuring biomass burning emission factors'. The new Figure 1 now also more clearly shows the setup. |
| Page 3 line 34: "Most atmospheric models account for photochemical processing, but not the chemical changes associated with the initial cooling of the smoke to ambient temperature."

 What types of chemical changes are associated with the initial cooling of the smoke? Please clarify. How can UAS-enable technique help to address this issue? | We meant changes that typically occur within minutes after emission. For our measurements, this impacts the condensation of volatile species, hygroscopic growth and the coating of black carbon particles by organic particles. Since the UAS measures seconds to minutes after emission, little photochemical alterations have taken place, which is particularly important when measuring short-lived emissions.

 We have revised the text to further clarify this, Line P4L1: "Most atmospheric models account for photochemical processing (e.g. oxidation of CO and $CH_4$), but not the chemical changes associated with the initial cooling of the smoke to ambient temperature (e.g. condensation of volatile species). EFs are therefore ideally measured in smoke that has already cooled to ambient temperature, but not yet undergone significant photochemical processing (Akagi et al., 2011). " |

| | |
|---|---|
| Section 2.1.2: A picture of the Mast and UAS setup would be helpful for the readers to gain a better idea of the measurement design (e.g., mast height, gas inlet of the mast, locations of equipment and sensors on mast, and locations of Tedlar bag and sensors on UAS). | We agree and have added a picture and short schematic overview of the proposed UAS. This is the new Figure 1 (first page of this document). |
| Page 5 lines 26-31: This paragraph is apparently too short, as this manuscript focuses on UAS-based measurement. The description regarding UAS sampling needs to be significantly improved. Although the authors mentioned that detailed methodology can be found in Vernooij et al. (2021), there are only two short paragraphs presented in that paper.

Questions to clarify:
a. Why DJI M100 was selected? What was the payload limit of DJI M100?

b. How was the Tadlar bag mounted to the UAS? How was gas sampling performed? Was a pump used during the sampling? Was it a Teflon pump? What was the flow rate of the sampling? How much air was collected in each bag? Was the flow rate recorded during the sampling? Was temperature recorded during the sampling for concentration correction? Was sampling quality control well performed (e.g., no leaking)? Samples were analyzed within 12 hours of sampling. How were the samples stored? Did it affect the concentration of the species inside the bag?

c. What were the weights of the AE51 and AM520? Given the payload, how long was each flight? Was the sampling inlet position on the top of UAS or below UAS? Were the data of AE51 and AM520 transmitted to the UAS controller at real time? Or they were stored in SD cards and retrieved after each flight?

d. Were UAS gas sampling and PM measurement influenced by the propellers? What were the influence lengths of the UAS-induced wind field above and below the UAS? Did it affect the mixing of biomass burning emissions and thus impact the estimation of emission factors? | We agree and have revised the text and added the new Figure:

"Figure 1 shows a schematic overview of the proposed measurement systems installed on the UAS (DJI Matrice 200). To prevent rotor-induced pressure alterations affecting aerosol mass concentrations, the inlets were extended using a carbonfibre tube to outside the rotor affected area. The gas-sampling system (530g) consists of an Arduino operated remote control and logging module, a 1.6 L min$^{-1}$ diaphragm gas pump (NMP 015, KNF), a four-way manifold connected to four 1L Tedlar bags and a carbonfibre cage (Fig 1., schematics in blue). This setup was based on an earlier setup deployed on a DJI Matrice 100 UAS described in Vernooij et al. (2020), but included a flushing mechanism to flush the extended inlet before sampling. The aerosol sampling system (Fig 1, schematics in black) contains two inlets. The first inlet is fitted with an inertial impactor (Personal Modular Impactor, SKC) followed by a 37mm quartz-fibre filter (Tissuquartz 2500QAT-UP, Merck) and a flow-controlled pump. The filtered air is used to dilute the stream coming from the second inlet using a 1:1 ratio to prevent saturation of the AM520 and limit the filter loading effect in the AE51. All tubing is polyurethane and kept as short as possible, whilst avoiding sharp corners. PM2.5 and eBC are continually logged in the AM520 and AE51 respectively. For each sample, the start time, end time, temperature, pressure and relative humidity at the UAS are logged. The transport time from the inlet to the measurement equipment is corrected for when computing EFs. The overall flight time of the system is roughly 15 minutes for each set of TB50 batteries, which is enough to fill 12-16 bags. The combined set-up equipped on a Matrice 200, takes collocated measurements of the $CO_2$, CO, $CH_4$ and $N_2O$ mixing ratio, as well as the PM$_{2.5}$ and eBC mass concentration, and compute the BB EFs of these species. While an earlier version of the UAS in Fig.1 (the DJI Matrice 100) was used for some of the experiments described in this study, this has insufficient payload capability for the combined system."

To clarify your individual questions: |

| | a) The DJI M100 was only used to carry the gas sampling system. The payload of the M100 is insufficient to carry both the aerosol and the gas sampling system. For the combined set-up we use a DJI Matrice 200 with a payload limit of 2.34kg |
| | b) We hope the added text (see above) clarifies these issues. |
| | c) The AE51 and AM520 respectively weigh 280 and 660 grams and each include their own battery. Both log the data (1 Hz) in their internal storage and the data is retrieved at the end of the fire.
Each flight is only a couple of minutes before the bags (35 sec each) need to be changed. During a fire we go through 14 DJI TB50 batteries, giving us over 2 hours of measurement time. Following similar questions from Reviewer #1 we have added this information in the text. |
| | d) In the Matrice 200 set-up which included the aerosol system, the inlet is kept away from the rotors. In the Matrice 100 set-up the inlet was in the middle of the UAS. However, the propellers should not affect the gas ratios. |
| Page 5 line 27: UAS sampling was conducted at an altitude of 15 m, similar as the sampling altitude at the mast. Why was 15 m selected for emission factor calculation? Please clarify. | We chose 15 meters because it is an altitude which is often high enough to safely fly the drone over the fire during intense late-dry season, but also low enough to still get elevated concentrations during weaker fires making for a better signal to noise ratio. We have added this to P6L22 of the revised manuscript.

While not quantitative, from our observations this seemed to be an altitude were both the characteristic white smoke from RSC and the black smoke from the flaming combustion mixed. |
| Page 6 line 12: "For the bag and mast measurements, we used the PM to CO ratio based on AM520 and CRDS measurements, with carbon accounting for 68% of the PM-mass (Reid et al., 2005a)."
This sentence is confusing. I understand that PM to CO ratio was calculated based on AM520 and CRDS. How was the PM carbon fraction calculated? Did you calculate the carbon fraction? Or it was obtained from Reid et al. (2005). If the fraction was obtained from the literature, why this value was selected? How representative is this value? | The AM520 outputs the particulate mass concentration. Therefore, the fraction of that mass represented by carbon is necessary to obtain the total carbon in PM, which is necessary for the carbon mass balance method used to calculate EFs.

In their analysis Reid et al. (2005) find that carbon accounts for ~50 to 70% of the mass of all particle emissions in all fires. In the savannas, the average is 68%, which is what we used for this study. Overall, the carbon in PM constitutes a small fraction of the total emitted carbon (~0.5-2%), so the effect of this assumption on the EFs is limited.

We added the following text to section 2.2:
"Overall, the carbon in PM and NMHC constitute respectively 0.5−2% and 0.4−3% of the total emitted carbon. Therefore, the uncertainty from the effect this assumption on the EFs of gaseous species is limited." |

| | |
|---|---|
| Page 7 lines 20-23 and Page 13 lines 3-5: as the authors mentioned, one of the reasons that AE51 is not as accurate as AE33, especially at low concentrations, is that AE51 operates at a much lower flow rate. One suggestion for future improvement can be increasing the sampling flow rate of AE51 using an external pump, as demonstrated by Wu et al. (Science of the Total Environment, 2021).

Reference: Wu et al., Vertical profiling of black carbon and ozone using a multicopter unmanned aerial vehicle (UAV) in urban Shenzhen of South China, Science of the Total Environment, 2021, 801, 149689. | We appreciate the suggestion and will take this into account for future work. The AE51 is deliberately set to a low flow rate, and without the use of an external pump could be increases from 50ml min$^{-1}$ to 200ml min$^{-1}$. Because of the high concentrations of BC in smoke (much higher than atmospheric profiling conducted by Wu et al., 2021), we followed the manufacturers guidelines and limited the flowrate to prevent loading effect non-linearity. Nonetheless, we believe doubling the flowrate may indeed improve the accuracy over less intense fires. Also, filter strips may be changed more frequently (several times per fire) to allow for higher flowrate while avoiding the loading effect. |
| Page 11 lines 5-6: "After the flaming phase ceased, mixing ratios and thus temporal varying EFs (green lines) for CO, CH$_4$ and PM$_{2.5}$ sharply rose."
What do you mean by "After the flaming phase ceased"? Do you mean after around 3 min in Figure 1? The emission factors of CO, CH$_4$, and PM2.5 increased sharply after 3 min in Figure 1. However, why no increases in emissions were observed? Instead, the emissions decreased significantly after 3 min. | The flaming phase is determined from sight and defined when the emissions from the flaming front have passed the mast, after ~2-3 minutes. We indeed mean that after this period, the relative emissions of CO, CH$_4$, and PM$_{2.5}$ increased, albeit that their absolute emissions diminished.

We rephrased the sentence to: 'After the emissions from the flaming front had passed the mast (~2-3 minutes), EFs (green lines) for CO, CH$_4$ and PM$_{2.5}$ sharply rose. Although the absolute emissions diminished, some emissions for these species persisted for the entire duration of the measurement.' |
| Page 16 line 32: "concentration. Cross-correlation of the UAV-mounted AM520 to five co-located AM520 modules revealed measurement errors of up to 20%."

Can the authors provide the figures showing the good correlation between the datasets collected from AM520 onboard UAV and those from AM520 on the mast? This can be good supporting information demonstrating the validity of UAS data. | Unfortunately, we don't have a direct comparison between UAS and mast aerosol measurements. We agree with the reviewer that this would be a very nice comparison, and hope to be able to do this in the future. We have therefore added this to the Recommendations and future improvements section: 'In the future, further tests of the set-up could be performed using additional inter-comparison of both aerosol and GHG EFs with mast measurements that include vertical velocity (e.g. FASS tower, Hao et al., 1996)' |
| The authors did a very good job discussing the assumptions and the uncertainties of this study. It would also be valuable to the readers if the authors can provide some recommendations (or improvements) for future studies using the UAS system for emission factor measurement. | Thank you, we added the following paragraph:

'4.5 Recommendations and future improvements

UAS payloads and lightweight sensors are continuously improving, meaning the UAS can in the future be equipped with more sophisticated sensors. The conversion of scattering parameters to particle mass may benefit from size-dependent CF. Although high concentration measurement may require some additional dilution, lightweight sensors like the Portable Optical Particle Spectrometer (POPS) (Mei et al., 2020) can measure particle size distribution. We also found that in fresh smoke, the contribution of BrC to the total absorption of BB particles was significant. Measurements at an additional short-wavelength band may therefore benefit absorption measurements. In the future, further tests of the set-up could be performed using additional inter-comparison of both aerosol and |

| | |
|---|---|
| | GHG EFs with mast measurements that include vertical velocity (e.g. FASS tower, Hao et al., 1996) as well as top-down approaches (e.g. van der Velde et al., 2020). ' |
| Technical corrections | Thank you for the corrections, the technical corrections were corrected accordingly in the revised document and are highlighted in the included 'track changes' document. |